# GAF is essential for zygotic genome activation and chromatin accessibility in the early *Drosophila* embryo

**Marissa M Gaskill†, Tyler J Gibson†, Elizabeth D Larson, Melissa M Harrison\***

Department of Biomolecular Chemistry, University of Wisconsin School of Medicine and Public Health, Madison, United States

**Abstract** Following fertilization, the genomes of the germ cells are reprogrammed to form the totipotent embryo. Pioneer transcription factors are essential for remodeling the chromatin and driving the initial wave of zygotic gene expression. In *Drosophila melanogaster*, the pioneer factor Zelda is essential for development through this dramatic period of reprogramming, known as the maternal-to-zygotic transition (MZT). However, it was unknown whether additional pioneer factors were required for this transition. We identified an additional maternally encoded factor required for development through the MZT, GAGA Factor (GAF). GAF is necessary to activate widespread zygotic transcription and to remodel the chromatin accessibility landscape. We demonstrated that Zelda preferentially controls expression of the earliest transcribed genes, while genes expressed during widespread activation are predominantly dependent on GAF. Thus, progression through the MZT requires coordination of multiple pioneer-like factors, and we propose that as development proceeds control is gradually transferred from Zelda to GAF.

**\*For correspondence:**
mharrison3@wisc.edu

†These authors contributed equally to this work

**Competing interests:** The authors declare that no competing interests exist.

## Introduction

Pronounced changes in cellular identity are driven by pioneer transcription factors that act at the top of gene regulatory networks. While nucleosomes present a barrier to the DNA binding of many transcription factors, pioneer factors can bind DNA in the context of nucleosomes. Pioneer-factor binding establishes accessible chromatin domains, which serve to recruit additional transcription factors that drive gene expression (*Zaret and Mango, 2016*; *Iwafuchi-Doi and Zaret, 2014*; *Zaret and Carroll, 2011*). These unique characteristics of pioneer factors enable them to facilitate widespread changes in cell identity. Nonetheless, cell-fate transitions often require a combination of pioneering transcription factors to act in concert to drive the necessary transcriptional programs. Indeed, the reprogramming of a specified cell type to an induced pluripotent stem cell requires a cocktail of transcription factors, of which Oct4, Sox2, and Klf4 function as pioneer factors (*Takahashi and Yamanaka, 2006*; *Takahashi et al., 2007*; *Chronis et al., 2017*; *Soufi et al., 2015*; *Soufi et al., 2012*). Despite the many examples of multiple pioneer factors functioning together to drive reprogramming, how these factors coordinate gene expression changes within the context of organismal development remains unclear.

Pioneer factors are also essential for the reprogramming that occurs in the early embryo. Following fertilization, specified germ cells must be rapidly and efficiently reprogrammed to generate a totipotent embryo capable of differentiating into all the cell types of the adult organism. This reprogramming is initially driven by mRNAs and proteins that are maternally deposited into the oocyte. During this time, the zygotic genome remains transcriptionally quiescent. Only after cells have been reprogrammed is the zygotic genome gradually activated. This maternal-to-zygotic transition (MZT) is broadly conserved among metazoans and essential for future development (*Schulz and Harrison, 2019*; *Vastenhouw et al., 2019*). Activators of the zygotic genome have been identified

**eLife digest** Most cells in an organism share the exact same genetic information, yet they still adopt distinct identities. This diversity emerges because only a selection of genes is switched on at any given time in a cell. Proteins that latch onto DNA control this specificity by activating certain genes at the right time. However, to perform this role they first need to physically access DNA: this can be difficult as the genetic information is tightly compacted so it can fit in a cell. A group of proteins can help to unpack the genome to uncover the genes that can then be accessed and activated. While these 'pioneer factors' can therefore shape the identity of a cell, much remains unknown about how they can work together to do so. For instance, the pioneer factor Zelda is essential in early fruit fly development, as it enables the genetic information of the egg and sperm to undergo dramatic reprogramming and generate a new organism. Yet, it was unclear whether additional helpers were required for this transition.

Using this animal system, Gaskill, Gibson et al. identified GAGA Factor as a protein which works with Zelda to open up and reprogram hundreds of different sections along the genome of fruit fly embryos. This tag-team effort started with Zelda being important initially to activate genes; regulation was then handed over for GAGA Factor to continue the process. Without either protein, the embryo died.

Getting a glimpse into early genetic events during fly development provides insights that are often applicable to other animals such as fish and mammals. Ultimately, this research may help scientists to understand how things can go wrong in human embryos.

in a number of species (zebrafish – Pou5f3, Sox19b, Nanog; mice – Dux, Nfy; humans – DUX4, OCT4; fruit flies – Zelda), and all share essential features of pioneer factors (*Schulz and Harrison, 2019*; *Vastenhouw et al., 2019*).

Early *Drosophila* development is characterized by 13, rapid, synchronous nuclear divisions. Zygotic transcription gradually becomes activated starting about the eighth nuclear division, and widespread transcription occurs at the 14th nuclear division when the division cycle slows and the nuclei are cellularized (*Schulz and Harrison, 2019*; *Vastenhouw et al., 2019*). The transcription factor Zelda (Zld) is required for transcription of hundreds of genes throughout zygotic genome activation (ZGA) and was the first identified global genomic activator (*Liang et al., 2008*; *Harrison et al., 2011*; *Nien et al., 2011*). Embryos lacking maternally encoded Zld die before completing the MZT (*Liang et al., 2008*; *Harrison et al., 2011*; *Nien et al., 2011*; *Fu et al., 2014*; *Staudt et al., 2006*). Zld has the defining features of a pioneer transcription factor: it binds to nucleosomal DNA (*McDaniel et al., 2019*), facilitates chromatin accessibility (*Schulz et al., 2015*; *Sun et al., 2015*) and this leads to subsequent binding of additional transcription factors (*Yáñez-Cuna et al., 2012*; *Xu et al., 2014*; *Foo et al., 2014*).

By contrast to the essential role for Zld in flies, no single global activator of zygotic transcription has been identified in other species. Instead, multiple transcription factors function together to activate zygotic transcription (*Schulz and Harrison, 2019*; *Vastenhouw et al., 2019*). Work from our lab and others has implicated additional factors in regulating reprogramming in the *Drosophila* embryo. Specifically, the enrichment of GA-dinucleotides in regions of the genome that remain accessible in the absence of Zld and at loci that gain accessibility late in ZGA, suggest that a protein that binds to these loci functions with Zld to define cis-regulatory regions during the initial stages of development (*Schulz et al., 2015*; *Sun et al., 2015*; *Blythe and Wieschaus, 2016*). Three proteins, GAGA factor (GAF), chromatin-linked adaptor for MSL proteins (CLAMP), and Pipsqueak (Psq) are known to bind to GA-dinucleotide repeats and are expressed in the early embryo, implicating one or all these proteins in reprogramming the embryonic transcriptome (*Rieder et al., 2017*; *Soruco et al., 2013*; *Kuzu et al., 2016*; *Biggin and Tjian, 1988*; *Bhat et al., 1996*; *Soeller et al., 1993*; *Lehmann et al., 1998*).

CLAMP was first identified based on its role in targeting the dosage compensation machinery, and it preferentially localizes to the X chromosome (*Larschan et al., 2012*; *Soruco et al., 2013*). Psq is essential for oogenesis (*Horowitz and Berg, 1996*), and it has been suggested to have a role as a repressor and in chromatin looping (*Gutierrez-Perez et al., 2019*; *Huang et al., 2002*). GAF,

encoded by the *Trithorax-like* (*Trl*) gene, has broad roles in transcriptional regulation, including functioning as a transcriptional activator (*Farkas et al., 1994*; *Bhat et al., 1996*), repressor (*Mishra et al., 2001*; *Busturia et al., 2001*; *Bernués et al., 2007*; *Horard et al., 2000*), and insulator (*Ohtsuki and Levine, 1998*; *Wolle et al., 2015*; *Kaye et al., 2017*). Through interactions with chromatin remodelers, GAF is instrumental in driving regions of accessible chromatin both at promoters and distal *cis*-regulatory regions (*Okada and Hirose, 1998*; *Tsukiyama et al., 1994*; *Xiao et al., 2001*; *Tsukiyama and Wu, 1995*; *Fuda et al., 2015*; *Judd et al., 2021*). Analysis of hypomorphic alleles has suggested an important function for GAF in the early embryo in both driving expression of *Ultrabiothorax* (*Ubx*), *Abdominal B* (*Abd-B*), *engrailed* (*en*), and *fushi tarazu* (*ftz*) and in maintaining normal embryonic development (*Bhat et al., 1996*; *Farkas et al., 1994*). Given these diverse functions for GAF, and that it shares many properties of a pioneer transcription factor, we sought to investigate whether it has a global role in reprogramming the zygotic genome for transcriptional activation.

Investigation of the role of GAF in the early embryo necessitated the development of a system to robustly eliminate GAF, which had not been possible since GAF is essential for maintenance of the maternal germline and is resistant to RNAi knockdown in the embryo (*Bhat et al., 1996*; *Bejarano and Busturia, 2004*; *Rieder et al., 2017*). For this purpose, we generated endogenously GFP-tagged GAF, which provided the essential functions of the untagged protein, and used the deGradFP system to deplete GFP-tagged GAF in early embryos expressing only the tagged construct (*Caussinus et al., 2012*). Using this system, we identified an essential function for GAF in driving chromatin accessibility and gene expression during the MZT. Thus, at least two pioneer-like transcription factors, Zld and GAF, must cooperate to reprogram the zygotic genome of *Drosophila* following fertilization.

## Results

### GAF binds the same loci throughout ZGA

To investigate the role of GAF during the MZT, we used Cas9-mediated genome engineering to tag the endogenous protein with super folder Green Fluorescent Protein (sfGFP) at either the N- or C-termini, sfGFP-GAF(N) and GAF-sfGFP(C), respectively (*Pédelacq et al., 2006*). There are two protein isoforms of GAF (*Benyajati et al., 1997*). Because the N-terminus of GAF is shared by both isoforms, the sfGFP tag on the N-terminus labels all GAF protein (*Figure 1—figure supplement 1A*). By contrast, the two reported isoforms differ in their C-termini, and thus the C-terminal sfGFP labels only the short isoform (*Figure 1—figure supplement 1B*). Whereas null mutants in *Trithorax-like* (*Trl*), the gene encoding GAF, are lethal (*Farkas et al., 1994*), both sfGFP-tagged lines are homozygous viable and fertile. Additionally, in embryos from both lines sfGFP-labeled GAF is localized to discrete nuclear puncta and is retained on the mitotic chromosomes in a pattern that recapitulates what has been previously described for GAF based on antibody staining in fixed embryos (*Figure 1A*, *Figure 1—figure supplement 1C*; *Raff et al., 1994*). Together, these data demonstrate that the sfGFP tag does not interfere with essential GAF function and localization.

To begin to elucidate the role of GAF during early embryogenesis, we determined the genomic regions occupied by GAF during the MZT. We hand sorted homozygous *GAF-sfGFP(C)* stage 3 (nuclear cycle (NC) 9) and stage 5 (NC14) embryos and performed chromatin immunoprecipitation coupled with high-throughput sequencing (ChIP-seq) using an anti-GFP antibody. While alternative splicing generates two GAF protein isoforms that differ in their C-terminal polyQ domain, in the early embryo only the short isoform is detectable (*Benyajati et al., 1997*). We confirmed the expression of the short isoform in the early embryo by blotting extract from 0 to 4 hr AEL (after egg laying) N- and C-terminally tagged GAF embryos with an anti-GFP antibody (*Figure 1—figure supplement 1D*). The long isoform was undetectable in extract from embryos of both sfGFP-tagged lines harvested 0–4 hr AEL, but was detectable at low levels in extract from *sfGFP-GAF(N)* embryos harvested 13–16 hr AEL. Thus, we conclude that during the MZT the C-terminally sfGFP-labeled short isoform comprises the overwhelming majority of GAF present in the *GAF-sfGFP(C)* embryos. We identified 3391 GAF peaks at stage 3 and 4175 GAF peaks at stage 5 (*Supplementary file 1*). To control for possible cross-reactivity with the anti-GFP antibody, we performed ChIP-seq on $w^{1118}$ stage 3 and stage 5 embryos in parallel. Since there are no GFP-tagged proteins in the $w^{1118}$

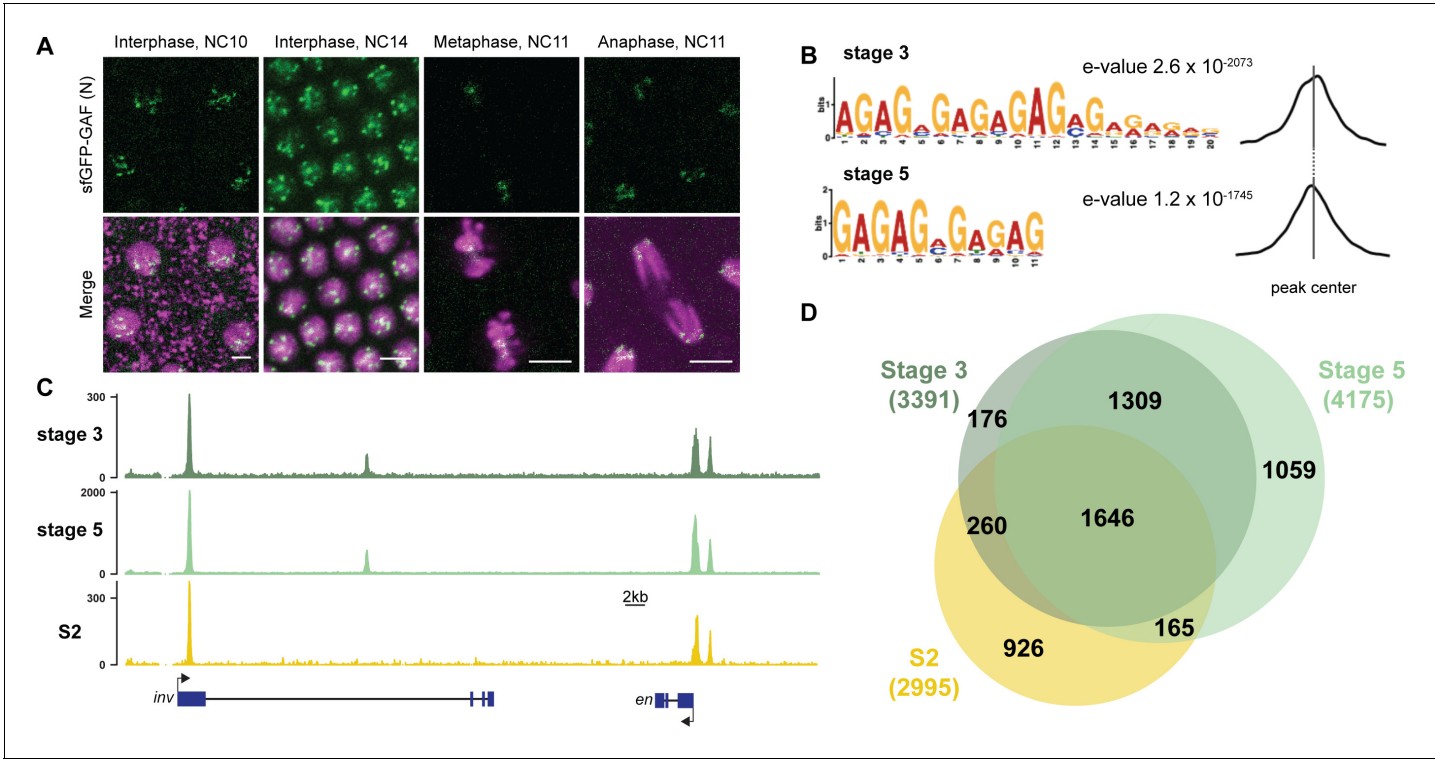

**Figure 1.** GAF binds thousands of loci throughout the MZT. (**A**) Images of His2Av-RFP; sfGFP-GAF(N) embryos at the nuclear cycles (NC) indicated above. sfGFP-GAF(N) localizes to puncta during interphase and is retained on chromosome during mitosis. His2AvRFP is shown in magenta. sfGFP-GAF (N) is shown in green. Scale bars, 5 μm. (**B**) Binding motif enrichment of GA-dinucleotide repeats at GAF peaks identified at stage 3 and stage 5 determined by MEME-suite (left). Distribution of the GA-repeat motif within peaks (right). Gray line indicates peak center. (**C**) Representative genome browser tracks of ChIP-seq peaks for GAF-sfGFP(C) from stage 3 and stage 5 embryos and GAF ChIP-seq from S2 cells (**Fuda et al., 2015**). (**D**) Venn diagram of the peak overlap for GAF as determined by ChIP-seq for GAF-sfGFP(C) from sorted stage 3 embryos and stage 5 embryos and by ChIP-seq for GAF from S2 cells (**Fuda et al., 2015**). Total number of peaks identified at each stage is indicated in parentheses. See also **Figure 1—figure supplements 1–3**.

The online version of this article includes the following figure supplement(s) for figure 1:

**Figure supplement 1.** N- and C- terminal sfGFP-tags label distinct GAF isoforms.

**Figure supplement 2.** GAF binds thousands of regions in the embyo and S2 cells.

**Figure supplement 3.** GAF has tissue-specific binding.

embryos, any peaks identified with the anti-GFP antibody would be the result of non-specific interactions and excluded from further analysis. No peaks were called in the $w^{1118}$ dataset for either stage, confirming the specificity of the peaks identified in the *GAF-sfGFP(C)* embryos (*Figure 1—figure supplement 2A,B*). Further supporting the specificity of the ChIP data, the canonical GA-rich GAF-binding motif was the most highly enriched sequence identified in ChIP peaks from both stages, and these motifs were centrally located in the peaks (*Figure 1B*). Peaks were identified in the regulatory regions of several previously identified GAF-target genes, including the *heat shock* promoter (*hsp70*), *even-skipped* (*eve*), *Krüppel* (*Kr*), *Ubx*, and *en* (*Figure 1C*; *Biggin and Tjian, 1988*; *Gilmour et al., 1989*; *Soeller et al., 1988*; *Lee et al., 1992*; *Read et al., 1990*; *Kerrigan et al., 1991*). Peaks were enriched at promoters, which fits with the previously defined role of GAF in establishing paused RNA polymerase (*Figure 1—figure supplement 2C*; *Lee et al., 2008*; *Fuda et al., 2015*; *Judd et al., 2021*).

There was a substantial degree of overlap between GAF peaks identified at both stage 3 and stage 5. A total of 2955 peaks were shared between the two time points, representing 87% of total stage 3 peaks and 71% of total stage 5 peaks (*Figure 1C,D*, *Figure 1—figure supplement 2D*). This demonstrates that GAF binding is established prior to widespread ZGA and remains relatively unchanged during early development, similar to what has been shown for Zld, the major activator of the zygotic genome (*Harrison et al., 2011*). We compared GAF-binding sites we identified in the

early embryo to previously identified GAF-bound regions in S2 cells, which are derived from 20 to 24 hr old embryos (*Fuda et al., 2015*). Despite the difference in cell-type and antibody used, 56% (1906) of stage 3 peaks and 43% (1811) of stage 5 peaks overlapped peaks identified in S2 cells (*Figure 1C,D*, *Figure 1—figure supplement 2D*). It was previously noted that GAF binding in 8–16 hr embryos was highly similar to GAF occupancy in the wing imaginal disc harvested from the larva (*Slattery et al., 2014*), and our data indicate that this binding is established early in development prior to activation of the zygotic genome. Nonetheless, peaks unique to each tissue likely represent GAF-binding events as they are centrally enriched for GA-rich GAF-binding motifs (*Figure 1—figure supplement 3*). Thus, while a majority of GAF-binding sites are maintained in both the embryo and cell culture, a subset of GAF-binding sites is likely tissue specific.

## Maternal GAF is required for embryogenesis and progression through the MZT

Early embryonic GAF is maternally deposited in the oocyte, and this maternally deposited mRNA can sustain development until the third instar larval stage (*Farkas et al., 1994*). To investigate the role of GAF during the MZT therefore necessitated a system to eliminate this maternally encoded GAF. RNAi failed to successfully knockdown GAF in the embryo (*Rieder et al., 2017*), and female germline clones cannot be generated as GAF is required for egg production (*Bhat et al., 1996*; *Bejarano and Busturia, 2004*). To overcome these challenges, we leveraged our N-terminal sfGFP-tagged allele and the previously developed deGradFP system to target knockdown at the protein level (*Caussinus et al., 2012*). The deGradFP system uses a genomically encoded F-box protein fused to a nanobody recognizing GFP, which recruits GFP-tagged proteins to a ubiquitin ligase complex. Ubiquitination of the GFP-tagged protein subsequently leads to efficient degradation by the proteasome. To adapt this system for efficient use in the early embryo, we generated transgenic flies in which the deGradFP nanobody fusion was driven by the *nanos (nos)* promoter for strong expression in the embryo 0–2 hr after fertilization (*Wang and Lehmann, 1991*). In embryos laid by *nos-deGradFP; sfGFP-GAF(N)* females all maternally encoded GAF protein is tagged with GFP and thus subject to degradation by the deGradFP nanobody fusion. These embryos are hereafter referred to as GAF$^{deGradFP}$.

We verified the efficiency of the deGradFP system by imaging living embryos in which nuclei were marked by His2Av-RFP. GAF$^{deGradFP}$ embryos lack the punctate, nuclear GFP signal identified in control embryos that do not carry the deGradFP nanobody fusion, indicating efficient depletion of sfGFP-GAF(N) (*Figure 2A*). This knockdown was robust, as we failed to identify any NC10 - 14 embryos with nuclear GFP signal, and none of the embryos carrying both the deGradFP nanobody and the sfGFP-tagged GAF hatched (*Figure 2B*). Based on live embryo imaging, the majority of embryos died prior to NC14, indicating that maternal GAF is essential for progression through the MZT. We identified a small number of GFP-expressing, gastrulating escapers. It is unclear if these embryos had an incomplete knockdown of maternally encoded sfGFP-GAF(N), or if a small percentage of embryos survived until gastrulation in the absence of GAF and that the GFP signal was the result of zygotic gene expression. Nonetheless, none of these embryos survived until hatching. Despite being able to maintain a strain homozygous for the N-terminal, sfGFP-tagged GAF, quantitative analysis revealed an effect on viability. Embryos homozygous for *sfGFP-GAF(N)* had only a 30% hatching rate (*Figure 2B*). By contrast, embryos homozygous for *GAF-sfGFP(C)* hatched at a rate of 66%. To determine whether the differences in hatching rates were due to differences in protein expression levels, we performed in vivo fluorescent quantification of the GFP signal in nuclei of *sfGFP-GAF(N)* and *GAF-sfGFP(C)* homozygous stage 5 embryos (2–2.5 hr AEL). There was no statistically significant difference in GFP signal between the genotypes (*Figure 2—figure supplement 1*), suggesting that the lower hatching rate for *sfGFP-GAF(N)* embryos is unlikely to be caused by reduced protein levels. Nonetheless, all future experiments controlled for the effect of the N-terminal tag by using *sfGFP-GAF(N)* homozygous embryos as paired controls with GAF$^{deGradFP}$ embryos.

Having identified a dramatic effect of eliminating maternally expressed GAF on early embryonic development, we used live imaging to investigate the developmental defects in our GAF$^{deGradFP}$ embryos. GAF$^{deGradFP}$ embryos in which nuclei were marked by a fluorescently labeled histone (His2Av-RFP) were imaged through several rounds of mitosis. We observed defects such as asynchronous mitosis, anaphase bridges, disordered nuclei, and nuclear dropout (*Figure 2C,D*; *Videos 1* and *2*). Nevertheless, embryos were able to complete several rounds of mitosis without GAF before

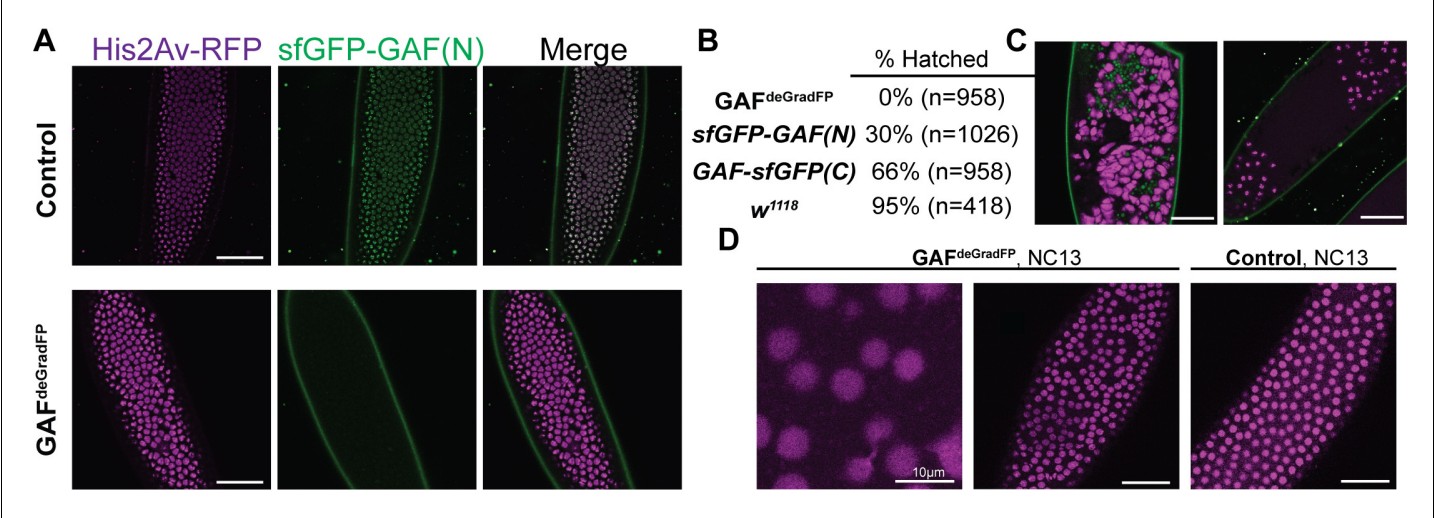

**Figure 2.** Embryos lacking maternal GAF die during early embryogenesis with nuclear and mitotic defects. (A) Images of control (maternal genotype: *His2Av-RFP; sfGFP-GAF(N)*) and GAF[deGradFP] (maternal genotype: *His2Av-RFP/nos-deGradFP; sfGFP-GAF(N)*) embryos at NC14, demonstrating loss of nuclear GFP signal specifically in GAF[deGradFP] embryos. His2Av-RFP marks the nuclei. (B) Hatching rates after > 24 hr. (C) Confocal images of His2Av-RFP in arrested/dying GAF[deGradFP] embryos with blocky nuclei, mitotic arrest, and nuclear fallout. (D) Confocal images of His2Av-RFP in NC13 control and GAF[deGradFP] embryos, showing disordered nuclei and anaphase bridges in GAF[deGradFP] embryos. Scale bars, 50 μm except where indicated. See also *Figure 2—figure supplement 1*.

The online version of this article includes the following figure supplement(s) for figure 2:

**Figure supplement 1.** sfGFP-GAF(N) and GAF-sfGF(C) are expressed at similar levels in the early embryo.

arresting. Nuclear defects became more pronounced as GAF[deGradFP] embryos developed and approached NC14. The arrested/dead embryos were often arrested in mitosis or had large, irregular nuclei (*Figure 2C*), similar to the nuclear 'supernovas' in GAF-deficient nuclei reported in *Bhat et al., 1996*. Our live imaging allowed us to detect an additional nuclear defect: that in the absence of maternal GAF nuclei become highly mobile and, in some cases, adopt a 'swirling' pattern (*Video 2*).

While it is unclear what causes in this phenotype, GAF is suggested to have a role in maintaining genome stability. Thus, loss of GAF may lead to activation of the DNA-damage response pathway, which can result in similar nuclear phenotypes (*Bhat et al., 1996*; *Sibon et al., 2000*; *Takada et al., 2003*). As a control, we imaged *sfGFP-GAF(N)* homozygous embryos through

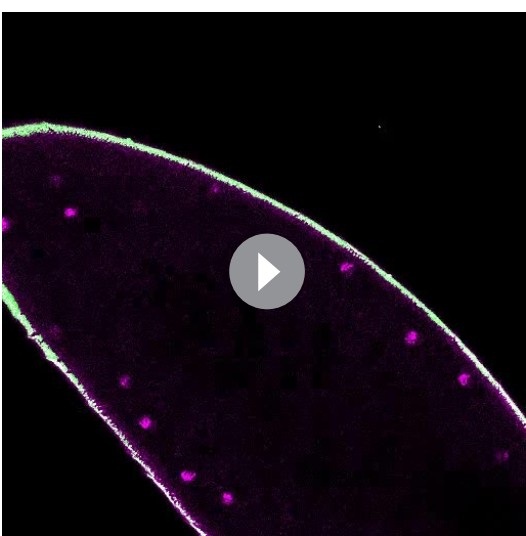

**Video 1.** Video of a GAF[deGradFP] embryo going through several rounds of mitosis prior to gastrulation. Nuclei are marked by His2Av-RFP.
https://elifesciences.org/articles/66668#video1

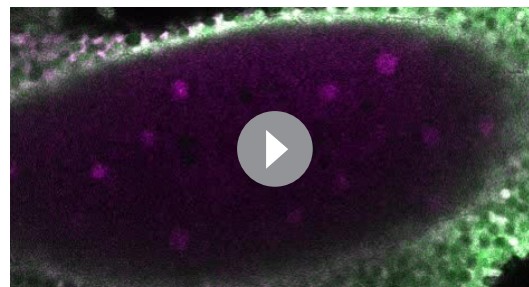

**Video 2.** Video of a severely disordered GAF[deGradFP] embryo going through several rounds of mitosis prior to gastrulation. Nuclei are marked by His2Av-RFP.
https://elifesciences.org/articles/66668#video2

several rounds of mitosis. These embryos proceeded normally through NC10-14, demonstrating that the mitotic defects in the GAF[deGradFP] embryos were caused by the absence of GAF and not the sfGFP tag (*Figure 2D*; *Video 3*). The defects in GAF[deGradFP] embryos are reminiscent of the nuclear-division defects previously reported for maternal depletion of *zld* (*Liang et al., 2008*; *Staudt et al., 2006*) and identify a fundamental role for GAF during the MZT.

## GAF is required for the activation of hundreds of zygotic genes

During the MZT, there is a dramatic change in the embryonic transcriptome as developmental control shifts from mother to offspring. Having demonstrated that maternal GAF was essential for development during the MZT, we investigated the role of GAF in regulating these transcriptional changes. We performed total-RNA seq on bulk collections of GAF[deGradFP] and control (*sfGFP-GAF (N)*) embryos harvested 2–2.5 hr AEL, during the beginning of NC14 when the widespread genome activation has initiated. Our replicates were reproducible (*Figure 3—figure supplement 1*), allowing us to identify 1452 genes that were misexpressed in GAF[deGradFP] embryos as compared to controls. Importantly, by using *sfGFP-GAF(N)* homozygous embryos as a control we have excluded from our analysis any genes misexpressed as a result of the sfGFP tag on GAF. Of the misexpressed genes 884 were down-regulated and 568 were up-regulated in the absence of GAF (*Figure 3A*, *Figure 3—figure supplement 2A*). The gene encoding GAF, *Trithorax-like*, was named because it was required for expression of the homeotic genes *Ubx* and *Abd-B* (*Farkas et al., 1994*). Our RNA-seq analysis identified 7 of the 8 *Drosophila* homeotic genes (*Ubx*, *Abd-B*, *adb-A*, *pb*, *Dfd*, *Scr*, and *Antp*) down-regulated in GAF[deGrad] embryos. Additionally, many of the gap genes, essential regulators of anterior-posterior patterning, are down-regulated: *giant* (*gt*), *knirps* (*kni*), *huckebein* (*hkb*), *Krüppel* (*Kr*), and *tailless* (*tll*) (*Supplementary file 2*). Gene ontology (GO)-term analysis showed down-regulated genes were enriched for functions in system development and developmental processes as would be expected for essential genes activated during ZGA (*Figure 3—figure supplement 2B*). GO-term analysis of the up-regulated genes showed weak enrichment for response to stimulus and metabolic processes (*Figure 3—figure supplement 2C*).

To determine whether GAF is functioning predominantly in transcriptional activation or repression, we used our stage 5 ChIP-seq data to determine the likely direct targets of GAF by assigning GAF-ChIP peaks to the nearest gene. We found that 45% (397) of the down-regulated genes were proximal to a GAF peak. By contrast, only 17% (99) of up-regulated genes are near GAF peaks, similar to the 15% of genes with unchanged expression levels that were proximal to GAF sites (*Figure 3A*). The significant enrichment for GAF-binding sites proximal to down-regulated genes as compared to up-regulated supports a role for GAF specifically in transcriptional activation ($p<2.2\times10^{-16}$, $\log_2$(odds ratio)=1.95, two-tailed Fisher's exact test). Genes activated late in NC14, during widespread genome activation, with a proximal GAF-binding site are significantly more down-regulated than genes activated at the same time point but lacking a GAF-binding site (*Figure 3—figure supplement 2D*). This analysis suggests that the down-regulated genes identified by RNA-seq are unlikely to be due to a general failure in activating zygotic gene expression, but rather specifically identify genes that depend on GAF for expression. Because zygotic gene expression is required for degradation of a subset of maternal mRNAs, if the genome is not activated, maternal transcripts are not properly degraded and therefore are increased in RNA-seq data (*Harrison et al., 2011*; *Hamm et al., 2017*; *Liang et al., 2008*). Indeed, down-regulated transcripts were enriched for zygotically expressed genes, and up-regulated transcripts were largely maternally

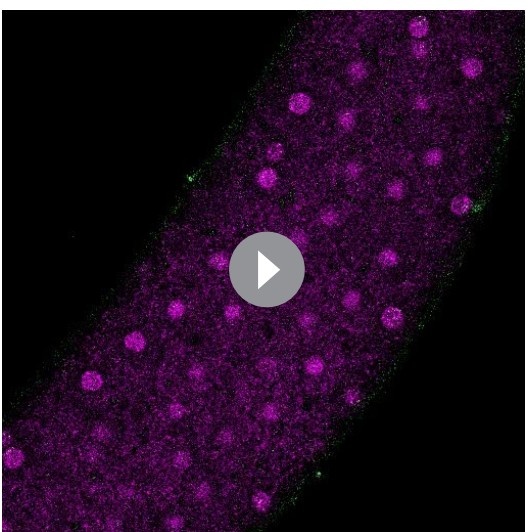

**Video 3.** Video of a control (*His2Av-RFP; sfGFP-GAF (N)*) embryo going through several rounds of mitosis prior to gastrulation. Nuclei are marked by His2Av-RFP.
https://elifesciences.org/articles/66668#video3

contributed ($p < 2.2 \times 10^{-16}$, $\log_2$(odds ratio)=4.21, two-tailed Fisher's exact test; *Figure 3B*). Thus, GAF is essential for transcriptional activation during the MZT and likely functions along with Zld to drive zygotic genome activation.

Previous data defined a role for the additional GA-dinucleotide binding protein, CLAMP, in activating zygotic gene expression (*Larschan et al., 2012*; *Rieder et al., 2017*; *Urban et al., 2017b*; *Soruco et al., 2013*). Therefore, we compared genes down-regulated in GAF$^{deGradFP}$ embryos to genes down-regulated in *clamp*-RNAi embryos 2–4 hr AEL (*Rieder et al., 2017*). A total of 174 genes are down-regulated in both datasets, comprising 19.7% of total down-regulated GAF targets and 50.1% of total down-regulated CLAMP targets (*Figure 3—figure supplement 3A*). While this demonstrates that a subset of genes requires both GAF and CLAMP for proper expression during ZGA, a majority of GAF-regulated genes only depend on GAF for expression, independent of CLAMP. GAF and CLAMP have similar, but not identical binding preferences (*Kaye et al., 2018*), which is reflected in their partially, but not completely overlapping genome occupancy (*Figure 3—figure supplement 3B*). These binding site differences likely explain their differential requirement during ZGA.

Having identified that GAF was required for ZGA, we wanted to ensure that the effects were not due to changes in the levels of Zld, the previously identified activator of the zygotic genome. Immunoblots for Zld on extract from GAF$^{deGradFP}$ and control (*sfGFP-GAF(N)*) 2–2.5 hr AEL (stage 5) embryos confirmed that Zld levels were consistent between extracts (*Figure 3—figure supplement 3C*). Therefore, the effects of GAF on genome activation are not due to a loss of Zld. Reciprocally, we performed immunoblots on 2–2.5 hr AEL embryo extract for sfGFP-GAF(N) in a background in which maternal *zld* was depleted using RNAi driven by *matα-GAL4-VP16* (*Sun et al., 2015*). We found that sfGFP-GAF(N) levels were unchanged upon *zld* knockdown compared to controls (*Figure 3—figure supplement 3D*), demonstrating that neither GAF nor Zld protein levels are affected by the knockdown of the other factor. Based on the roles for Zld and GAF in activating the zygotic genome, we investigated whether these proteins were required to activate distinct or overlapping target genes. We compared genes down-regulated in GAF$^{deGradFP}$ embryos to genes down-regulated when Zld was inactivated optogenetically throughout zygotic genome activation (NC10-14) (*McDaniel et al., 2019*). We identified 135 genes down-regulated in both datasets, comprising 42% of the total number of down-regulated genes dependent on Zld and 15% of the total down-regulated genes dependent on GAF (*Figure 3C*). An even lower degree of overlap is observed when only direct targets are considered with 49 down-regulated targets shared between Zld and GAF, which have 232 and 397 direct targets, respectively. By contrast only 29 up-regulated genes were shared between the two datasets (*Figure 3—figure supplement 3E*). Genes that required both factors for activation include the gap genes previously mentioned (*gt, kni, hkb, Kr, tll*) as well as genes involved in cellular blastoderm formation such as *slow as molasses* (*slam*) and *bottleneck* (*bnk*). While Zld and GAF share some targets, they each are required for expression of hundreds of individual genes.

Activation of the zygotic genome is a gradual process that initiates with transcription of a small number of genes around NC8. Transcripts can therefore be divided into categories based on the timing of their initial transcription (*Li et al., 2014*; *Lott et al., 2011*). Previous data suggested that while Zld was required for activation of genes throughout the MZT, early genes were particularly sensitive to loss of Zld and that GAF might be functioning later (*Schulz et al., 2015*; *Blythe and Wieschaus, 2016*). To determine when GAF-dependent genes were expressed during ZGA, we took all the genes that could be classified based on their timing of activation during the MZT (as determined in *Li et al., 2014*) and divided them based on their dependence on Zld and GAF for activation: those down-regulated in both GAF$^{deGradFP}$ and Zld$^{CRY2}$ embryos (Both), those down-regulated only in Zld$^{CRY2}$ embryos (Zld$^{CRY2}$), and those down-regulated only in GAF$^{deGradFP}$ embryos (GAF$^{deGradFP}$) (*Figure 3D*). Genes activated by Zld, both those regulated by Zld alone (Zld$^{CRY2}$) and those regulated by both Zld and GAF (Both) were enriched for genes expressed early (NC10-11) and Mid (NC12-13). By contrast, 73% of genes activated by GAF alone (GAF$^{deGradFP}$) were expressed either late (early NC14) or later (late NC14) ($p = 1.3 \times 10^{-14}$, $\log_2$(odds ratio)=3.13, two-tailed Fisher's exact test). This analysis supports a model in which GAF and Zld are essential activators for the initial wave of ZGA and that GAF has an additional role, independent of Zld, in activating widespread transcription during NC14.

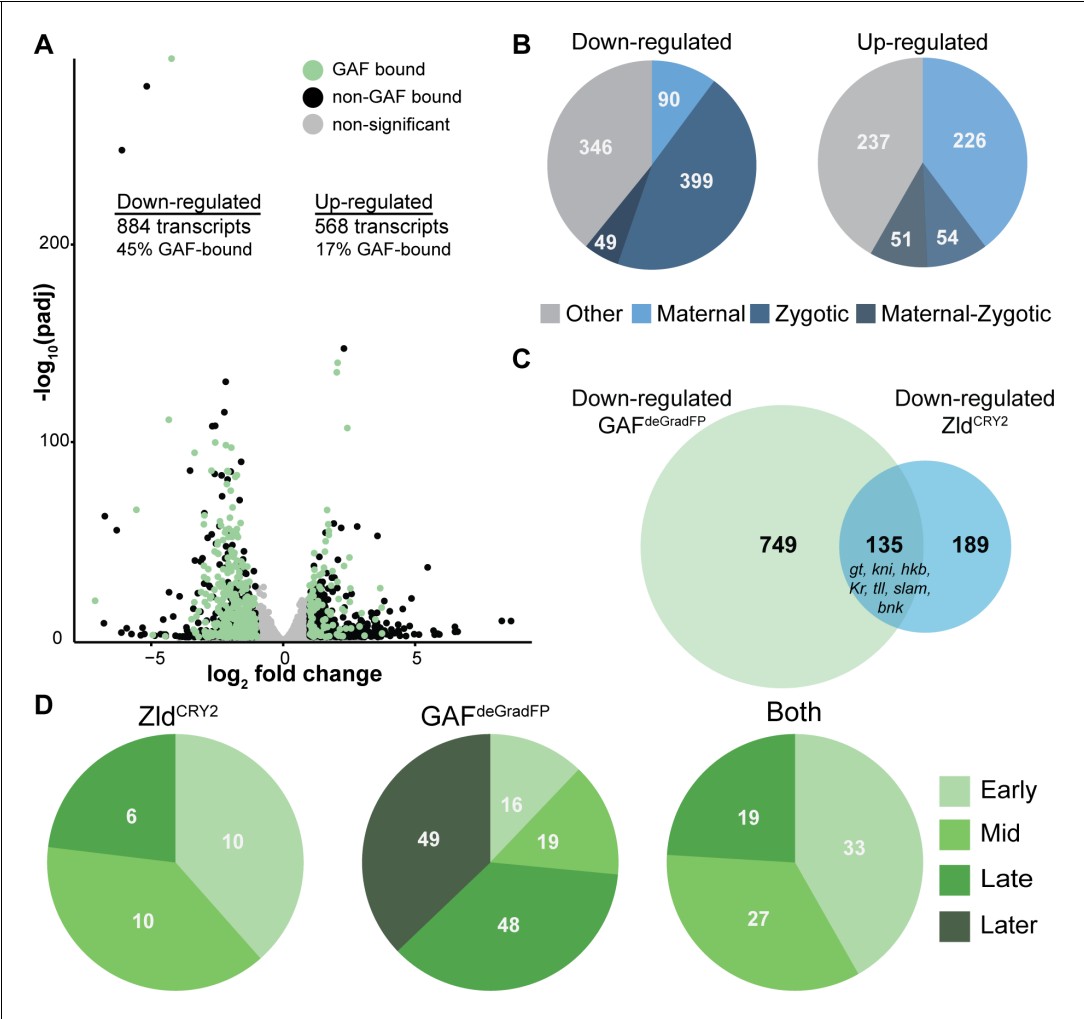

**Figure 3.** GAF is required for zygotic genome activation. (A) Volcano plot of transcripts mis-expressed in GAF[deGradFP] embryos as compared to *sfGFP-GAF(N)* controls. Stage 5 GAF-sfGFP(C) ChIP-seq was used to identify GAF-bound target genes. (B) The percentage of up-regulated and down-regulated transcripts in GAF[deGradFP] embryos classified as maternal, zygotic, or maternal-zygotic based on *Lott et al., 2011*. (C) Overlap of down-regulated embryonic transcripts in the absence of GAF or Zld activity ($p<2.2\times10^{-16}$, $\log_2$(odds ratio)=3.16, two-tailed Fisher's exact test). Down-regulated genes for Zld are from *McDaniel et al., 2019*. (D) Transcripts down-regulated in both GAF[deGradFP] and Zld[CRY2] embryos or in either condition alone were classified based on temporal expression during ZGA (*Li et al., 2014*). Early = NC10-11, Mid = NC12-13, Late = early NC14, Later = late NC14. Only genes that were assigned to one of the four classes are shown. See also *Figure 3—figure supplements 1–3*.

The online version of this article includes the following figure supplement(s) for figure 3:

**Figure supplement 1.** RNA-seq replicates are reproducible.

**Figure supplement 2.** RNA-seq identifies genes mis-regulated in GAF[deGradFP] embryos.

**Figure supplement 3.** GAF regulates genes distinct from Zld and CLAMP.

## The majority of GAF and Zld-binding sites are occupied independently of the other factor

To further delineate the relationship between Zld- and GAF-mediated transcriptional activation during the MZT, we determined whether GAF binding was dependent on Zld. Not only are a substantial subset of genes dependent on both Zld and GAF for wild-type levels of expression (*Figure 3C*), but 42% of GAF-binding sites at stage 5 are also occupied by Zld (*Harrison et al., 2011*; *Figure 4A,B*). The GAF peaks that are co-bound with Zld include many of the strongest GAF peaks identified at stage 5 (*Figure 4—figure supplement 1A*). Zld is known to facilitate the binding of multiple different transcription factors (Twist, Dorsal, and Bicoid), likely by forming dynamic subnuclear hubs (*Yáñez-Cuna et al., 2012*; *Xu et al., 2014*; *Foo et al., 2014*; *Mir et al., 2018*; *Dufourt et al., 2018*;

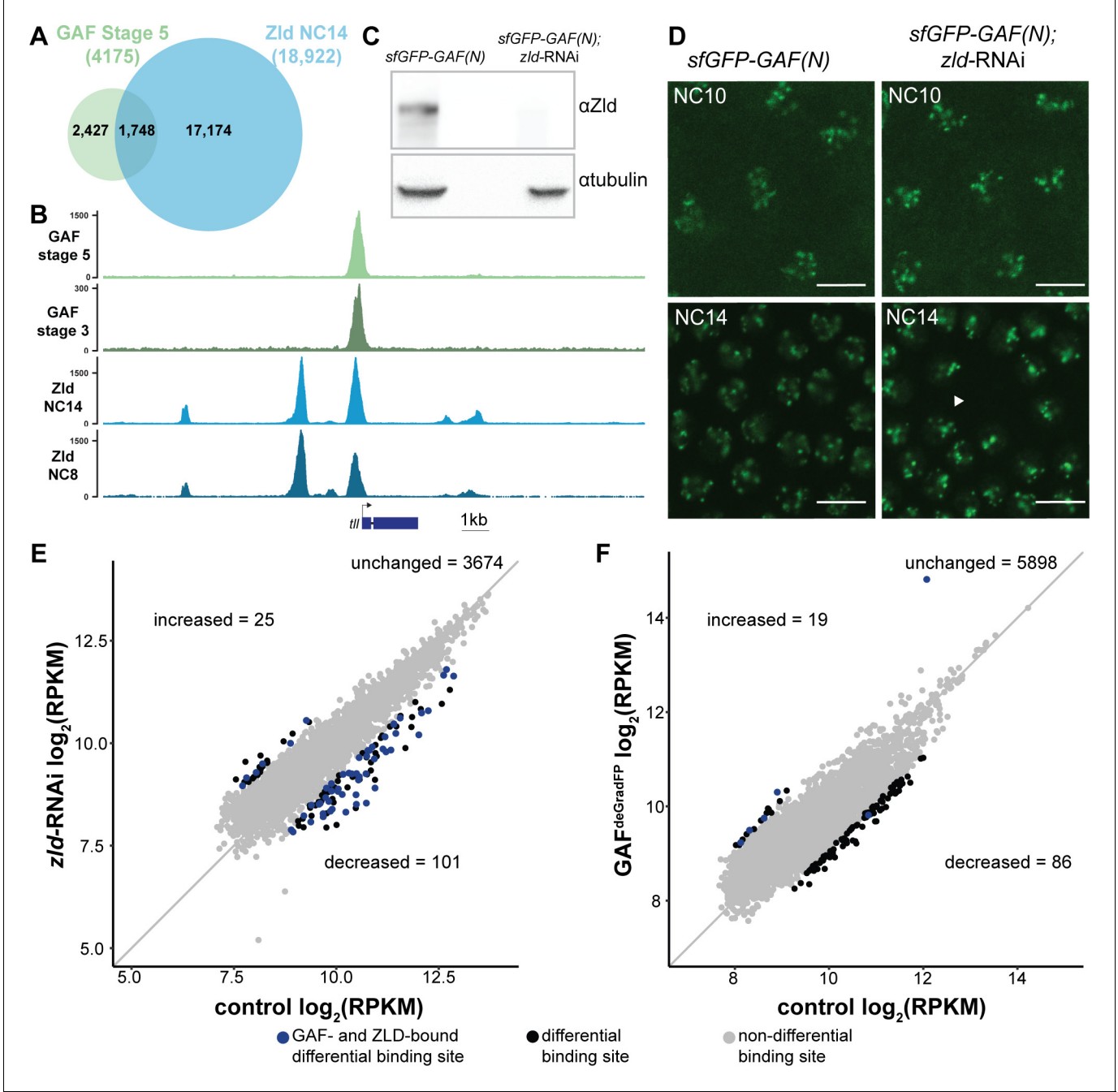

**Figure 4.** At the majority of loci, GAF and Zld bind chromatin independently. (A) Overlap of Zld- and GAF-binding sites determined by GAF-sfGFP(C) stage 5 ChIP-seq and Zld NC14 ChIP-seq (*Harrison et al., 2011*). (B) Representative genome browser tracks of Zld and GAF ChIP-seq peaks at the *tailless* locus. (C) Immunoblot for Zld on embryo extracts from *zld*-RNAi; *sfGFP-GAF(N)* and *sfGFP-GAF(N)* control embryos harvested 2–3 hr AEL. Tubulin was used as a loading control. (D) Images of *zld*-RNAi; *sfGFP-GAF(N)* embryos and *sfGFP-GAF(N)* control embryos at NC10 and NC14 as marked. Arrowhead shows nuclear dropout, a phenotype indicative of *zld* loss-of-function. Scale bar, 10 μm. (E) Correlation between log$_2$(RPKM) of ChIP peaks for GAF from *GAF-sfGFP(N)* stage 5 embryos (control) and *zld*-RNAi; *sfGFP-GAF(N)* embryos (*zld*-RNAi). Color highlights significantly changed peaks (adjusted p-value<0.05, fold change > 2) and those that are bound by both GAF and Zld as indicated below. (F) Correlation between log$_2$(RPKM) of ChIP peaks for Zld from *sfGFP-GAF(N)* embryos (control) and GAF$^{deGradFP}$ embryos fixed 2–2.5 hr AEL. Color highlights significantly changed peaks (adjusted p-value<0.05, fold change > 2) and those that are bound by both GAF and Zld as indicated below. See also *Figure 4—figure supplements 1–2*.

The online version of this article includes the following figure supplement(s) for figure 4:

**Figure supplement 1.** GAF and Zld bind to shared and unique regions of the genome.

**Figure supplement 2.** Independent chromatin binding by GAF and Zld.

*Yamada et al., 2019*). Indeed, 19% of Dorsal sites depend on Zld for occupancy (*Sun et al., 2015*). We therefore examined GAF localization upon depletion of maternal *zld* using RNAi driven in the maternal germline by *matα*-GAL4-VP16 in a background containing sfGFP-GAF(N) (*Sun et al., 2015*). Immunoblot confirmed a nearly complete knockdown of Zld (*Figure 4C*), and we verified that RNAi-treated embryos failed to hatch. To assess the depletion of Zld on the subnuclear localization of GAF, we imaged *sfGFP-GAF(N); zld*-RNAi and control (*sfGFP-GAF(N)*) embryos using identical acquisition settings. We observed no difference in puncta formation of GAF at the beginning of ZGA (NC10) or late ZGA (NC14) (*Figure 4D*). We conclude that Zld is not required for GAF to form subnuclear puncta during the MZT.

To more specifically determine the impact of loss of Zld on GAF chromatin occupancy, we performed ChIP-seq with an anti-GFP antibody on embryos expressing *zld*-RNAi and sfGFP-GAF(N) along with paired *sfGFP-GAF(N)* controls at 2–2.5 hr AEL (stage 5). Mouse H3.3-GFP chromatin was used as a spike-in to normalize for immunoprecipitation efficiency between samples. In control *sfGFP-GAF(N)* embryos, we identified 6373 peaks, and these largely overlapped with the peaks identified in the *GAF-sfGFP(C)* embryos; 91% of the 4175 peaks identified in the *GAF-sfGFP(C)* embryos overlap with the peaks identified for the N-terminally tagged GAF (*Figure 4—figure supplement 1B*). This high degree of overlap indicates that our ChIP-seq experiments identified a robust set of high-confidence GAF-bound regions. To determine whether GAF requires Zld for binding, we analyzed GAF occupancy upon *zld* knockdown at this set of 3800 high-confidence peaks. The majority (3674) of these high-confidence GAF peaks were maintained in the *zld*-RNAi background and were bound at roughly equivalent levels when normalized to the spike-in control (*Figure 4E* and *Figure 4—figure supplement 2A*). Using DESeq2, we identified 126 GAF-bound regions that were significantly different between the *sfGFP-GAF(N)* controls and the *zld*-RNAi embryos (*Figure 4E*): 101 sites that were decreased and 25 sites that were increased.

To assess the impact of the loss of GAF on Zld binding, we completed the reciprocal experiment and performed ChIP-seq for Zld on GAF[deGradFP] and control homozygous (*sfGFP-GAF(N)*) embryos at 2–2.5 hr AEL (stage 5). We identified a set of high-confidence Zld peaks, by overlapping the Zld peaks from our control (*sfGFP-GAF(N)*) embryos with previously published data for Zld at NC14 (*Harrison et al., 2011*). Similar to GAF binding when Zld is depleted, the majority of the 6003 high-confidence Zld peaks were maintained when GAF was depleted (*Figure 4F*). However, in the GAF[deGradFP] ChIP data we observed a global reduction in Zld peak heights as compared to the control (*Figure 4—figure supplement 2B*). Therefore, it is possible that GAF knockdown causes a global reduction in Zld binding. However, technical, rather than biological differences, may account for this overall decrease (see Materials and methods). Among the high-confidence Zld peaks, DESeq2 identified 105 Zld-binding sites that differed in occupancy between GAF[deGradFP] embryos and controls: 86 sites decreased and 19 sites increased.

Because of the limited number of significantly different peaks identified upon removal of either Zld (3.3% of GAF peaks changed) or GAF (1.7% of Zld peaks changed), we sought to determine whether there were global changes in the distribution of these factors. For this purpose, we ranked peaks in each dataset based on the number of reads per kilobase per million mapped reads (RPKM) and determined if the peak ranks were correlated between the mutant and control. We identified a high degree of correlation in peak rank for both comparisons (Pearson correlation, r = 0.93 for GAF ChIP in *sfGFP-GAF(N)* and *sfGFP-GAF(N); zld-RNAi* embryos and r = 0.88 for Zld ChIP in *sfGFP-GAF(N)* and GAF[deGradFP] embryos; *Figure 4—figure supplement 2C,D*). Thus, the overall distribution of binding sites is maintained for each factor in the absence of the other.

At some loci, pioneer factors have been shown to function together to stabilize binding (*Donaghey et al., 2018*; *Chronis et al., 2017*). If either GAF or Zld stabilized genomic occupancy of the other factor, we would expect to identify a loss of binding specifically at loci that are co-bound by both factors. We identified a set of GAF and Zld co-occupied sites by overlapping our high-confidence GAF peaks and high-confidence Zld peaks. Peaks with at least 100 bp overlap were considered to be shared and used for our set of co-bound regions. To test likely direct effects of GAF on Zld occupancy, we determined the number regions that showed significant changes in Zld binding in GAF[deGradFP] embryos that were bound by GAF. One out of the 86 regions at which Zld binding decreased in the GAF knockdown and 5 out of 19 that increased overlapped with regions bound by GAF (*Figure 4F*). Contrary to our expectation if GAF was directly promoting Zld occupancy, there was no enrichment for co-bound regions and those that lost Zld binding. We investigated the

reciprocal effect of Zld on GAF occupancy by determining the number regions that showed significant changes in GAF binding in *zld*-RNAi embryos that were also bound by Zld. Fifty-one of the 101 regions that had decreased GAF binding were also bound by Zld, suggesting a potential direct effect of Zld on GAF occupancy at these regions. By contrast, only 6 of the 25 regions that had increased GAF binding were also bound by Zld (*Figure 4E*). GAF-binding regions that were decreased in the absence of Zld were significantly enriched for co-bound sites as compared to those GAF-binding sites that were maintained upon Zld depletion (p=$4.8\times10^{-5}$, log$_2$(odds ratio)=1.19, two-tailed Fisher's exact test). Based on the enrichment for Zld binding at these GAF-bound decreased sites, we hypothesized that Zld might be pioneering chromatin accessibility at these regions to promote GAF binding. Indeed, these regions were enriched for loci that depend on Zld for chromatin accessibility (p=$2.2\times10^{-16}$, log$_2$(odds ratio)=4.07, two-tailed Fisher's exact test): 38 of the 51 GAF and Zld co-bound sites at which GAF requires Zld for occupancy also require Zld for accessibility (*Hannon et al., 2017*). Thus, at this limited set of regions the pioneer factor Zld may function to establish accessible chromatin that is necessary to promote robust GAF binding. Nonetheless, the majority of Zld- and GAF-binding sites are occupied in the absence of the other factor.

## GAF is essential for accessibility at hundreds of loci during ZGA

GAF interacts with chromatin remodelers and is required to maintain chromatin accessibility in tissue culture (*Okada and Hirose, 1998*; *Tsukiyama et al., 1994*; *Tsukiyama and Wu, 1995*; *Xiao et al., 2001*; *Judd et al., 2021*; *Fuda et al., 2015*). Furthermore, GAF-binding motifs are enriched at regions of open chromatin that are established at NC12 and NC13 and that are bound by Zld but do not depend on Zld for accessibility (*Schulz et al., 2015*; *Sun et al., 2015*; *Blythe and Wieschaus, 2016*). To directly test the function of GAF in determining accessible chromatin domains during the MZT, we performed the assay for transposase-accessible chromatin (ATAC)-seq on six replicates of single GAF$^{deGradFP}$ and control (*sfGFP-GAF(N)*) embryos harvested 2–2.5 hr AEL. His2AV-RFP signal for each embryo was visually inspected prior to performing ATAC-seq to ensure that embryos did not have grossly distorted nuclear morphology. In contrast to our control, replicates from the GAF$^{deGradFP}$ embryos showed higher variability (*Figure 5—figure supplement 1A*), suggesting that developmental defects caused by the lack of GAF might have lowered our ability to detect subtle changes in accessibility. Nonetheless, we identified 1523 regions with significant changes in accessibility in GAF$^{deGradFP}$ embryos as compared to controls; 607 regions lost accessibility and 916 regions gained accessibility (*Figure 5A*, *Figure 5—figure supplement 1B,C*). Sites that lost accessibility were among the regions with the highest ATAC-signal (*Figure 5—figure supplement 1C*). 32% (197) of the regions that lost accessibility were significantly enriched for GAF binding, as determined by stage 5 *GAF-sfGFP(C)* ChIP-seq, when compared to open regions with no change in accessibility (p=$2.2\times10^{-16}$, log$_2$(odds ratio)=2.4, two-tailed Fisher's exact test, *Figure 5—figure supplement 1D*). The enrichment for GAF-binding sites in those regions that depend on GAF for accessibility suggests that GAF may directly drive accessibility at these regions. Therefore, we focused our analysis on the subset of regions that depend on GAF for accessibility, which we define as those that lost accessibility in the absence of GAF. Consistent with the enrichment of GAF-binding sites in promoters, 45% of all regions that depend on GAF for accessibility were in promoters. Enrichment for promoters was even larger in those regions that were bound by GAF and dependent on GAF for accessibility (*Figure 5B*). By contrast, regions that gain accessibility in the absence of GAF are not enriched for promoters, supporting that this is likely an indirect effect of GAF knockdown (*Figure 5B*). *Ubx*, a previously identified embryonic GAF-target gene, is an example of a gene that requires GAF for chromatin accessibility at the promoter (*Figure 5C*; *Farkas et al., 1994*). *Ubx* similarly requires GAF occupancy for gene expression as the transcript was significantly down-regulated in GAF$^{deGradFP}$ embryos (*Figure 5C*). Thus, GAF binding at the *Ubx* promoter is required for both chromatin accessibility and gene expression. Together, our data demonstrate that GAF is required for chromatin accessibility at hundreds of loci during the MZT, and that this activity occurs preferentially at promoters.

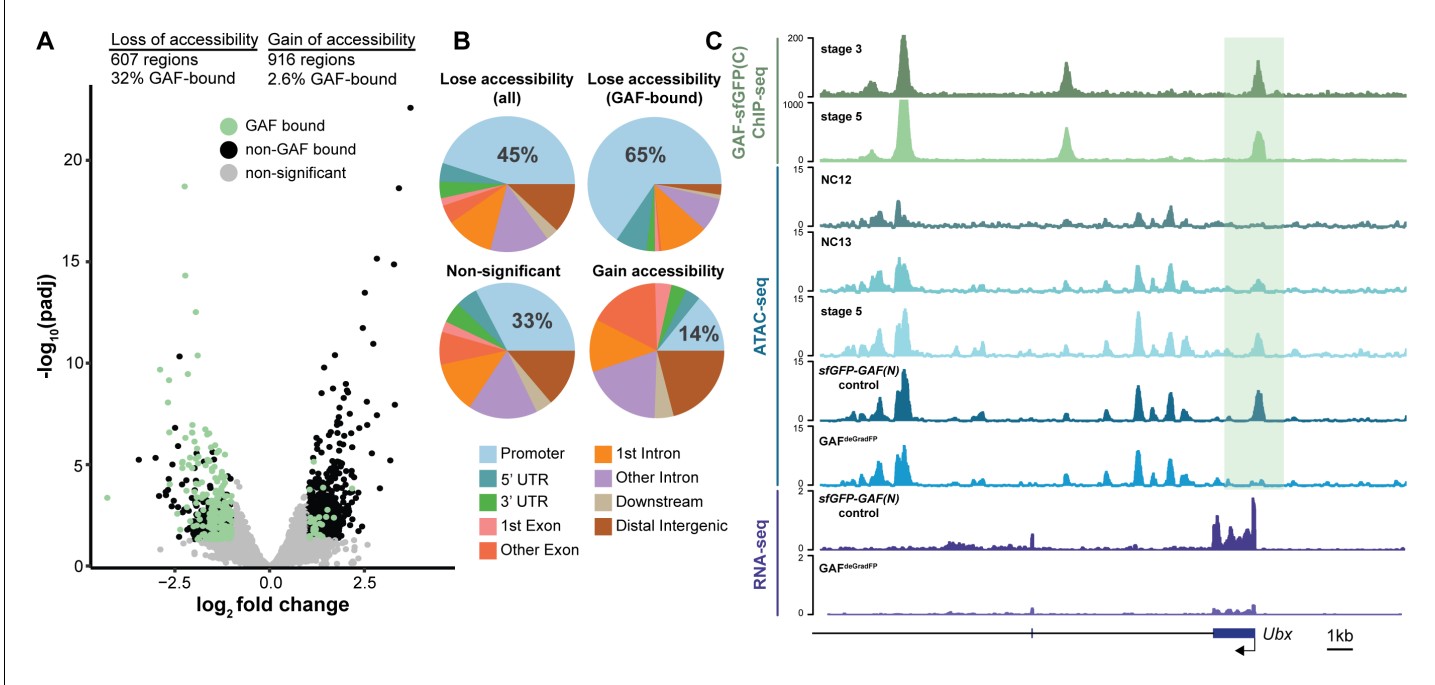

**Figure 5.** GAF is required for chromatin accessibility. (**A**) Volcano plot of regions that change in accessibility in GAF[deGradFP] embryos as compared to *sfGFP-GAF(N)* controls, stage 5 GAF-sfGFP(C) ChIP-seq was used to identify GAF-bound target regions. (**B**) Genomic distribution of all regions that lose accessibility (Lose accessibility (all)), regions that lose accessibility and are GAF-bound (Lose accessibility (GAF-bound)), regions that did not change significantly in accessibility (Non-significant), and regions that gain accessibility (Gain accessibility)). (**C**) Genome browser tracks of GAF-sfGFP(C) ChIP-seq at stage 3 and stage 5, ATAC-seq on wild-type embryos at NC12, NC13, and stage 5 along with control (sfGFP-GAF(N)) and GAF[deGradFP] embryos, and RNA-seq from control (sfGFP-GAF(N)) and GAF[deGradFP] embryos. NC12 and NC13 ATAC-seq data is from *Blythe and Wieschaus, 2016*. ATAC-seq data for stage 5 embryos is from *Nevil et al., 2020*. Region highlighted in green indicates the GAF-dependent, GAF-bound *Ubx* promoter. See also *Figure 5—figure supplement 1*.

The online version of this article includes the following figure supplement(s) for figure 5:

**Figure supplement 1.** GAF is required for chromatin accessibility.

## GAF and Zld are individually required for chromatin accessibility at distinct regions

Previous work from our lab and others suggested that during the MZT, GAF may be responsible for maintaining chromatin accessibility at Zld-bound regions in the absence of Zld (*Schulz et al., 2015*; *Moshe and Kaplan, 2017*; *Sun et al., 2015*). To more broadly investigate how GAF and Zld shape chromatin accessibility during early development, we focused on regions co-occupied by both factors as identified by ChIP-seq (this work and *Harrison et al., 2011*). We compared our single embryo ATAC-seq data to ATAC-seq data for wild-type NC14 embryos and NC14 embryos lacking maternal *zld* (*zld⁻*) (*Figure 6A*; *Hannon et al., 2017*). Only seven of the 1192 regions co-bound by both Zld and GAF decreased in accessibility upon removal of either factor. By contrast, 104 regions require GAF for accessibility and 190 require Zld. Regions that require GAF for accessibility had a higher average GAF ChIP-seq peak height than regions that require Zld. Similarly, the Zld ChIP-seq signal was higher in regions where Zld is necessary for accessibility. Thus, both GAF and Zld are individually required for accessibility at distinct genomic regions co-occupied by both factors, and this requirement is correlated with occupancy as reflected in ChIP-seq peak height (*Figure 6A*). GAF is also required for accessibility at 83 additional regions at which GAF is bound without Zld (*Figure 6—figure supplement 1*). The majority of sites bound by both Zld and GAF (891) did not change in chromatin accessibility when either factor was removed, suggesting that at these locations GAF and Zld may function redundantly to facilitate chromatin accessibility or that other factors were sufficient to maintain accessibility at these sites. Thus, during the MZT GAF and Zld are individually required for

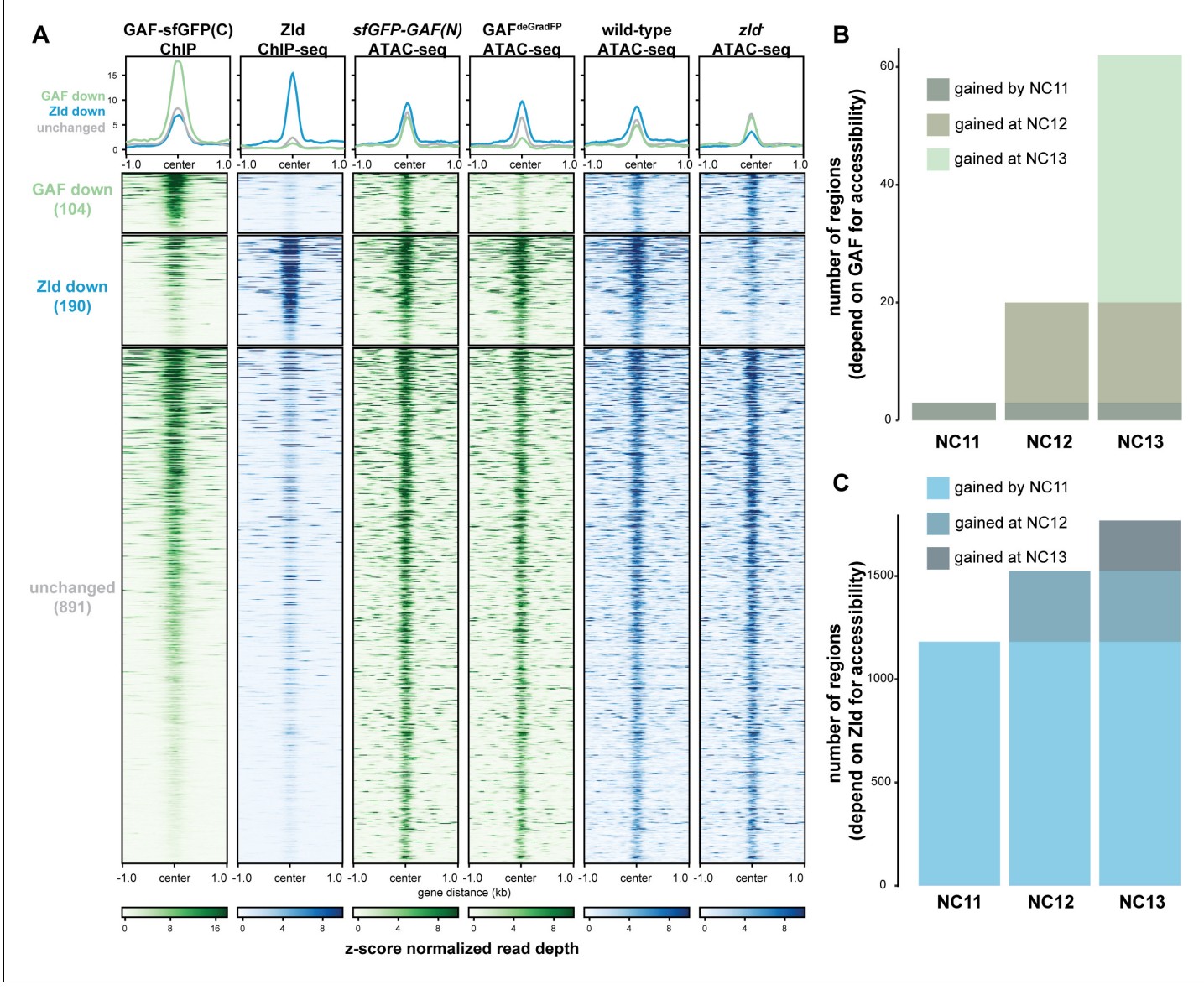

**Figure 6.** GAF and Zld independently shape chromatin accessibility over the MZT. (A) Heatmaps of ChIP-seq and ATAC-seq data, as indicated above, for regions bound by both GAF and Zld and subdivided based on the change of accessibility in the absence of either factor. (B) Number of regions that depend on GAF for accessibility and are bound by GAF that are accessible at NC11, NC12, and NC13. (C) Number of regions that depend on Zld for accessibility and are bound by Zld that are accessible at NC11, NC12, and NC13. NC11, NC12, and NC13 data are from *Blythe and Wieschaus, 2016*. Zld ChIP-seq data are from *Harrison et al., 2011*. ATAC-seq data from *zld* germline clones (*zld⁻*) are from *Hannon et al., 2017*. Total number of regions accessible at NC11 = 3084, NC12 = 6487, and NC13 = 9824. See also *Figure 6—figure supplements 1–2*.

The online version of this article includes the following figure supplement(s) for figure 6:

**Figure supplement 1.** A subset of regions bound by GAF, and not Zld, depend on GAF for accessibility.

**Figure supplement 2.** Regions that gain accessibility late during the MZT are accessible in GAF$^{deGradFP}$ embryos used for ATAC-seq.

chromatin accessibility at distinct co-bound regions. At very few co-bound regions are both factors individually required.

Previous work demonstrated that the Zld-binding motif is enriched at accessible regions that are established by NC11. By contrast, GAF-binding motifs are enriched at regions that dynamically gain chromatin accessibility later during the MZT at NC12 or NC13 (*Blythe and Wieschaus, 2016*). If Zld preferentially drives chromatin accessibility early in the MZT and GAF is preferentially required later, we would expect that regions that require GAF for accessibility would be enriched for regions that

gain accessibility at NC12 and NC13. To test this prediction, we compared our ATAC-seq data to ATAC-seq on embryos precisely staged by nuclear cycle (*Blythe and Wieschaus, 2016*). Of the GAF-bound, GAF dependent accessible regions that were identified in the staged ATAC-seq data, three were open by NC11, 17 were newly opened at NC12, and 42 were newly opened at NC13 (*Figure 6B*, *Figure 6—figure supplement 2A*), demonstrating that GAF-bound, GAF-dependent regions are enriched for regions that dynamically open at NC12 or NC13 as compared to regions that are already accessible at NC11 (p=$5.7\times10^{-7}$, $\log_2$(odds ratio)=3.2, two-tailed Fisher's exact test). Indeed, the *Ubx* promoter dynamically gained accessibility at NC13 (*Figure 5C*). Confirming that the loss of accessibility observed at late-opening regions was not due to differences in staging between the control (*sfGFP-GAF(N)*) and GAF$^{deGradFP}$ embryos, the vast majority of sites that gained accessibility at NC13 (2452) were unchanged in accessibility in the GAF$^{deGradFP}$ embryos (*Figure 6— figure supplement 2*). In contrast to GAF, Zld-bound, Zld-dependent accessible regions were enriched for regions already accessible at NC11: 1182 were open by NC11, 343 were newly opened at NC12, and 244 were newly opened at NC13 (p<$2.2\times10^{-16}$, $\log_2$(odds ratio)=2.7, two-tailed Fisher's exact test) (*Figure 6C*). Together, these analyses show that GAF is preferentially required for chromatin accessibility at regions that gain accessibility over the MZT, while Zld is required for accessibility at regions that are open early.

The preferential requirement for GAF-mediated accessibility during NC12 and NC13 allowed us to test whether GAF binding preceded the establishment of chromatin accessibility at these regions or whether this accessibility was co-incident with GAF binding. We determined whether the 42 regions that depended on GAF for accessibility and that gained accessibility at NC13 were already bound by GAF at stage 3 (NC9) or whether these regions were among those that were only occupied by GAF at stage 5 (NC14). All 42 of these regions were bound by GAF at stage 3 and stage 5, demonstrating that GAF occupancy at these regions precedes accessibility and supports a role for GAF as a pioneer factor at these regions.

## Discussion

Through a combination of Cas9-genome engineering and the deGradFP system, we have depleted maternally encoded GAF and demonstrated that GAF is required for progression through the MZT. Along with Zld, GAF is broadly required for activation of the zygotic genome and for shaping chromatin accessibility. During the major wave of ZGA, when thousands of genes are transcribed, Zld and GAF are largely independently required for both transcriptional activation and chromatin accessibility (*Figure 7*). Thus, in *Drosophila*, as in mice, zebrafish, and humans, transcriptional activation during the MZT is driven by multiple factors with pioneering characteristics (*Schulz and Harrison, 2019*; *Vastenhouw et al., 2019*). Together our system has enabled us to begin to determine how these two powerful factors collaborate to ensure the rapid, efficient transition from specified germ cells to a pluripotent cell population.

### Maternally encoded GAF is essential for development beyond the MZT

Using the deGradFP system, we demonstrated that maternally encoded GAF is required for embryogenesis. In contrast to zygotic GAF null embryos, which survive until the third instar larval stage, GAF$^{deGradFP}$ embryos do not hatch (*Farkas et al., 1994*). Our imaging showed that depletion of GAF in the early embryo resulted in severe defects, including asynchronous mitosis, anaphase bridges, nuclear dropout, and high nuclear mobility (*Figure 2C,D*, *Videos 1* and *2*). Similar defects are seen when *zld* is maternally depleted from embryos (*Liang et al., 2008*; *Staudt et al., 2006*) and like embryos lacking maternal *zld*, GAF$^{deGradFP}$ embryos died before the completion of the MZT. Distinct from embryos lacking maternal *zld*, a subset of GAF$^{deGradFP}$ embryos display more intense defects. We identified defects in a subset of GAF$^{deGradFP}$ embryos as early as NC10, which is consistent with our genomics data indicating that GAF is required for proper expression of some of the earliest genes expressed during ZGA (*Figure 3D*). It is possible that some of the phenotypic defects observed in GAF$^{deGradFP}$ embryos are the result of GAF knockdown in the female germline. However, GAF null female germline clones are unable to produce eggs (*Bejarano and Busturia, 2004*), which stands in contrast to the large number of GAF$^{deGradFP}$ embryos produced by females expressing sfGFP-GAF(N) and the deGradFP nanobody fusion. Together these phenotypic differences between null mutants and our deGrad-based knockdown along with our genomics data identifying

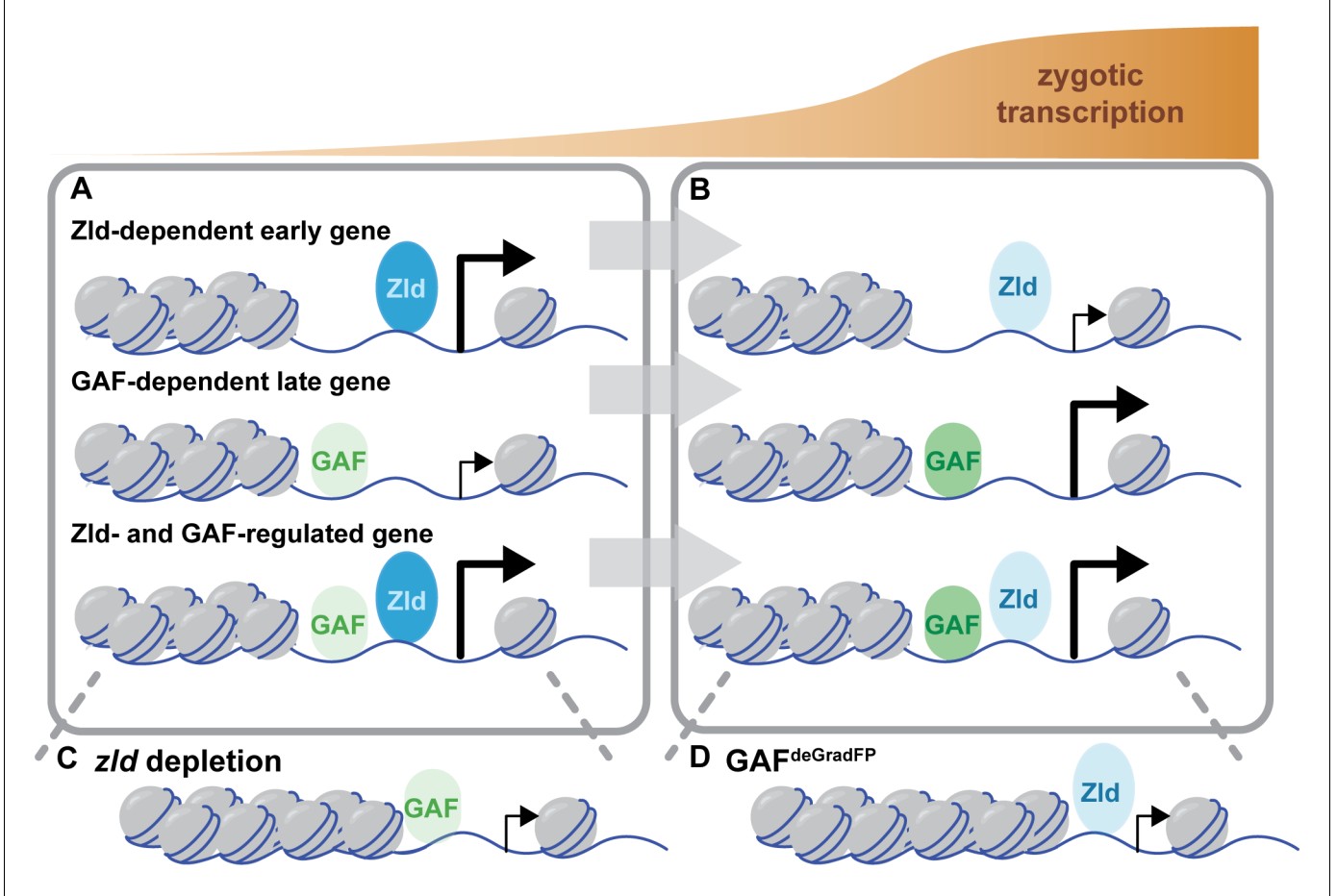

**Figure 7.** Zld and GAF independently regulate embryonic reprogramming. (**A**) During the minor wave of ZGA (NC10-13), Zld is the predominant factor required for driving expression of genes bound by Zld alone and genes bound by both Zld and GAF. (**B**) As the genome is more broadly activated during NC14, GAF becomes the major factor in driving zygotic transcription. (**C**) Early in the MZT, Zld is required for chromatin accessibility at many regions co-bound by GAF and Zld. When *zld* is depleted, accessibility is lost at a subset of regions, but GAF remains bound at the majority of sites. (**D**) Late during the MZT, GAF is required for chromatin accessibility at many Zld-bound regions. In GAF^deGradFP embryos accessibility is lost at a subset of sites, but Zld remains bound.

misregulation of previously identified embryonic GAF-targets in the GAF^deGradFP embryos demonstrate an essential function for GAF in the early embryo.

Live imaging of GFP-tagged, endogenously encoded GAF demonstrated that GAF is mitotically retained in small foci. This is similar to what was reported for antibody staining on fixed embryos, which showed GAF localized at pericentric heterochromatin regions of GA-rich satellite repeats (*Raff et al., 1994*; *Platero et al., 1998*). Because this prior imaging necessitated fixing embryos, it was unclear if mitotic retention of GAF was required for mitosis. Our system enabled us to determine that in the absence of GAF nuclei can undergo several rounds of mitosis albeit with noticeable defects (*Videos 1* and *2*). We conclude that GAF is not strictly required for progression through mitosis. However, the nuclear defects observed in GAF^deGradFP embryos support the model that GAF is broadly required for nuclear division and chromosome stability, in addition to its role in transcriptional activation during ZGA (*Bhat et al., 1996*). Our imaging also identified high nuclear mobility in GAF^deGradFP embryos as compared to control embryos; nuclei of a subset of embryos showed a dramatic 'swirling' pattern of movement (*Video 2*). This defect is potentially due a DNA damage response that inactivates centrosomes to promote nuclear fallout (*Sibon et al., 2000*; *Takada et al., 2003*), or general disorder in the cytoskeletal network that is responsible for nuclear migration and division in the syncytial embryo (*Sullivan and Theurkauf, 1995*). Altogether, our phenotypic analysis of GAF^deGradFP embryos shows for the first time that maternal GAF is required for progression

through the MZT, and suggests GAF has an early, global role in nuclear division and chromosome stability.

In addition to GAF and Zld, the transcription factor CLAMP is expressed in the early embryo and functions in chromatin accessibility and transcriptional activation (*Rieder et al., 2017*; *Soruco et al., 2013*; *Urban et al., 2017a*; *Urban et al., 2017b*; *Rieder et al., 2019*). While CLAMP, like GAF, binds GA-dinucleotide repeats, the two proteins preferentially bind to slightly different GA-repeats (*Kaye et al., 2018*). GAF and CLAMP can compete for binding sites in vitro, and, in cell culture, when one factor is knocked down the occupancy of the other increases, suggesting that CLAMP and GAF compete for a subset of binding sites and may have partial functional redundancy (*Kaye et al., 2018*). We demonstrate that GAF, like CLAMP, is essential in the early embryo, indicating that these two GA-dinucleotide-binding proteins cannot completely compensate for each other in vivo during early development (*Rieder et al., 2017*). Furthermore, our sequencing analysis identified that a majority of genes that require GAF for expression are distinct from those that are regulated by CLAMP. Thus, while GAF and CLAMP may have some overlapping functions, they are independently required to regulate embryonic development during the MZT.

## GAF is necessary for widespread zygotic genome activation and chromatin accessibility

GAF is a multi-purpose transcription factor with known roles in transcriptional regulation at promoters and enhancers as well as additional suggested roles in high-order chromatin structure. Our analysis showed that during the MZT GAF acts largely as an activator, directly binding and activating hundreds of zygotic transcripts during ZGA (*Figure 3*). We identified thousands of regions bound by GAF throughout the MZT, and these regions were preferentially associated with genes whose transcription decreased when maternally encoded GAF was degraded. This function may be driven, in part, through GAF-mediated chromatin accessibility as we identified hundreds of regions that depend on GAF for accessibility (*Figure 5*). This activity, in both transcriptional activation and mediating open chromatin, is similar to Zld, the only previously identified essential activator of the zygotic genome in *Drosophila,* and is shared with genome activators in other species, such as Pou5f3 and Nanog (zebrafish) and Dux (mammals) (*Schulz and Harrison, 2019*; *Vastenhouw et al., 2019*). Our data support a pioneer-factor like role for GAF in this process as we demonstrated that GAF is already bound early in the MZT to regions that require GAF for accessibility later. Thus, GAF occupancy precedes GAF-mediated accessibility at a subset of sites.

In addition to a direct role in mediating chromatin accessibility through the recruitment of chromatin remodelers (*Okada and Hirose, 1998*; *Tsukiyama et al., 1994*; *Tsukiyama and Wu, 1995*; *Xiao et al., 2001*; *Judd et al., 2021*; *Fuda et al., 2015*), GAF may also indirectly affect chromatin accessibility through a role in shaping three-dimensional chromatin structure. Sixty-seven percent of the regions that lost chromatin accessibility in the absence of GAF did not overlap a GAF-binding site as identified by ChIP-seq. Thus, at these regions GAF may function indirectly through the ability to facilitate enhancer-promoter loops (*Mahmoudi et al., 2002*; *Melnikova et al., 2004*; *Petrascheck et al., 2005*). In addition, GAF-binding motifs are enriched at TAD boundaries which form during the MZT (*Hug et al., 2017*), and GAF binding is enriched at Polycomb group dependent repressive loops that form following NC14 (*Ogiyama et al., 2018*). Further investigation is necessary to determine the role of GAF in establishing three-dimensional chromatin architecture in the early embryo.

## GAF and Zld are uniquely required to reprogram the zygotic genome

Reprogramming in culture requires a cocktail of transcription factors that possess pioneer factor activity (*Takahashi and Yamanaka, 2006*; *Soufi et al., 2012*; *Soufi et al., 2015*). Similarly, in zebrafish and mice multiple pioneering factors are required for the rapid and efficient reprogramming that occurs during the MZT (*Schulz and Harrison, 2019*; *Vastenhouw et al., 2019*). Here we have shown that, in addition to the essential pioneer factor Zld, GAF is a pioneer-like factor required for both gene expression and chromatin accessibility in the early *Drosophila* embryo. By analyzing the individual contributions of these two factors, we have begun to elucidate the different mechanisms by which multiple pioneer factors can drive dramatic changes in cell identity.

While some pioneer factors work together to stabilize their interaction on chromatin (*Chronis et al., 2017*; *Donaghey et al., 2018*; *Liu and Kraus, 2017*; *Swinstead et al., 2016*) our data support primarily independent genome occupancy by Zld and GAF. Our reciprocal ChIP-seq data demonstrated that at regions bound by GAF and Zld, binding of each factor was largely retained in the absence of the other (*Figure 4*). Analysis of chromatin accessibility further supports these independent roles. We would predict that if Zld binding were lost in the absence of GAF or vice versa that at regions occupied by both factors either Zld or GAF individually would be required for accessibility. However, we identified only seven regions that lost accessibility in the absence of either Zld or GAF. By contrast, we identified more than one hundred regions that were individually dependent on each factor alone, and 891 regions that did not change in accessibility upon loss of either Zld or GAF. These data support independent roles for Zld and GAF in establishing or maintaining accessibility. Furthermore, we propose that because each factor can retain genome occupancy in the absence of the other that this may explain the 891 regions that remain accessible in the absence of either Zld or GAF: in the absence of Zld, GAF may be able to maintain accessibility and vice versa. However, until we investigate the chromatin landscape upon the removal of both Zld and GAF we cannot rule out that at these regions other factors may be instrumental in maintaining accessibility. Indeed, Duan et al. identify an essential role for the GA-dinucleotide-binding protein CLAMP in directing Zld binding and promoting chromatin accessibility (*Duan et al., 2020*).

At most loci, our data support independent binding for GAF and Zld. Nonetheless, we cannot eliminate a possible global role for GAF on Zld occupancy as the Zld ChIP-seq signal is considerably lower in the GAF$^{deGradFP}$ embryos as compared to controls. We would predict that if GAF is directly functioning to stabilize Zld binding then this effect would be more evident at regions where both GAF and Zld bind, but this is not what we observed. Nonetheless, because of the proposed role of GAF in regulating three-dimensional chromatin architecture, GAF may be globally required for robust Zld occupancy. Furthermore, at a small subset of loci our data support a pioneering role for Zld in promoting GAF binding. In this experiment, the endogenously tagged GAF allowed us to use antibodies directed against the GFP epitope for ChIP. The use of the epitope tag enabled us to robustly control for immunoprecipitation efficiency by using another GFP-tagged protein as a spike-in control for the immunoprecipitation of GAF. In this manner, we identified 101 GAF-bound regions that decreased upon Zld knockdown. Of these, 51 were Zld-bound regions and 38 of these depend on Zld for chromatin accessibility. Thus, at a small subset of sites the pioneering function of Zld may be required to stabilize GAF binding.

We propose that there is a handoff between Zld and GAF pioneer-like activity as the embryo progresses through the MZT: Zld functions primarily at the initiation of zygotic genome activation and GAF functions primarily later during the major wave of zygotic transcription. We identified that genes regulated by Zld are enriched for those activated during NC10-13, while genes that depend on GAF are enriched for those that initiate expression during NC14. Similarly, we determined that the majority of regions that require Zld for accessibility are already accessible at NC11 while those that require GAF are enriched for regions that gain accessibility at NC12 and NC13. This is supported by prior analysis that showed that the Zld-binding motif is enriched at regions that are already accessible at NC11 and the GAF-binding motif is enriched at regions that dynamically gain accessibility later during the MZT (*Blythe and Wieschaus, 2016*). Based on this evidence, we propose that there is a gradual handoff in the control of transcriptional activation and chromatin remodeling from Zld to GAF as the MZT progresses (*Figure 7*). During the initial stages of the MZT, the nuclear division cycle is an incredibly rapid series of synthesis and mitotic phases. At NC14, this cycle slows, and these different dynamics likely influence the mechanisms by which accessibility can be established. While it is unclear how Zld establishes accessibility, GAF interacts with chromatin remodelers. It is possible that the activities of these complexes may have a more substantial impact once the division cycle slows.

Together our data support the requirement for at least two pioneer-like transcription factors, Zld and GAF, to sequentially reprogram the zygotic genome following fertilization and allow for future embryonic development. It is likely that there are additional factors that function along with Zld and GAF to define accessible *cis*-regulatory regions and drive genome activation. Future studies will enable more detailed mechanistic insights into how multiple pioneering factors work together to reshape the transcriptional landscape and transform cell fate in the early embryo.

# Materials and methods

## Key resources table

| Reagent type (species) or resource | Designation | Source or reference | Identifiers | Additional information |
|---|---|---|---|---|
| Genetic reagent (*Drosophila melanogaster*) | $w^{1118}$ | Bloomington *Drosophila* Stock Center | BDSC:3605; FLYB:FBal0018186; RRID:BDSC_3605 | |
| Genetic reagent (*D. melanogaster*) | *His2AV-RFP (II)* | Bloomington *Drosophila* Stock Center | BDSC:23651; FLYB:FBti0077845; RRID:BDSC_23651 | |
| Genetic reagent (*D. melanogaster*) | *mat-α-GAL4-VP16* | Bloomington *Drosophila* Stock Center | BDSC:7062; FLYB:FBti0016915; RRID:BDSC_7062 | |
| Genetic reagent (*D. melanogaster*) | *UAS-shRNA-zld* | *Sun et al., 2015* DOI:10.1101/gr.192542.115 | | |
| Genetic reagent (*D. melanogaster*) | *sfGFP-GAF(N)* | This paper | | Cas9 edited allele |
| Genetic reagent (*D. melanogaster*) | *GAF-sfGFP(C)* | This paper | | Cas9 edited allele |
| Genetic reagent (*D. melanogaster*) | *nos-deGradFP* | This paper | | Transgenic insertion into *PBac{yellow[+]-attP-3B} VK00037* docking site (BDSC:9752) (FLYB: FBti0076455) *NSlmb-vhhGFP4* amplified from BDSC:58740 |
| Antibody | Anti-GFP (rabbit polyclonal) | Abcam | Cat# ab290 | ChIP (6 µg) WB (1:2000) |
| Antibody | Anti-Zld (rabbit polyclonal) | *Harrison et al., 2010* DOI:10.1016/j.ydbio.2010.06.026 | | ChIP (8 µg) WB (1:750) |
| Antibody | Anti-alpha tubulin (mouse monoclonal) | Sigma-Aldrich | Cat# T6199 | WB (1:5000) |
| Antibody | Anti-rabbit IgG-HRP (Goat, secondary) | Bio-Rad | Cat#1706515 | WB (1:3000) |
| Antibody | Anti-mouse IgG-HRP (Goat, secondary) | Bio-Rad | Cat#1706516 | WB (1:3000) |
| cell line *Mus musculus* | H3.3-GFP | This paper | | Cell line maintained in the lab of Peter Lewis |
| software, algorithm | R | http://www.R-project.org | | |
| software, algorithm | bowtie 2 v2.3.5 | *Langmead and Salzberg, 2012* | | |
| software, algorithm | Samtools v1.11 | http://www.htslib.org/ | | |
| software, algorithm | MACS v2 | *Zhang et al., 2008* | | |
| software, algorithm | GenomicRanges R package | *Lawrence et al., 2013* | | |
| software, algorithm | DeepTools | *Ramírez et al., 2016* | | |
| software, algorithm | MEME-suite | *Bailey et al., 2009* | | |
| software, algorithm | Gviz R package | *Hahne and Ivanek, 2016* | | ' |
| software, algorithm | Subread (v1.6.4) | *Liao et al., 2014* | | |
| software, algorithm | DESeq2 R package | *Love et al., 2014* | | |
| software, algorithm | HISAT v2.1.0 | *Kim et al., 2015* | | |
| software, algorithm | NGMerge | *Gaspar, 2018* | | |

### *Drosophila* strains and genetics

All stocks were grown on molasses food at 25°C. Fly strains used in this study: $w^{1118}$, *His2Av-RFP (II)* (Bloomington *Drosophila* Stock Center (BDSC) #23651), *mat-α-GAL4-VP16* (BDSC #7062), *UAS-shRNA-zld* (Sun et al., 2015). *sfGFP-GAF(N)* and *GAF-sfGFP(C)* mutant alleles were generated using Cas9-mediated genome engineering (see below).

  *nos-deGradFP (II)* transgenic flies were made by PhiC31 integrase-mediated transgenesis into the *PBac{yellow[+]-attP-3B}VK00037* docking site (BDSC #9752) by BestGene Inc. The sequence for *NSlmb-vhhGFP4* was obtained from Caussinus et al., 2012 and amplified from genomic material from *UASp-Nslmb.vhhGFP4* (BDSC #58740). The *NSlmb-vhhGFP4* sequence was cloned using Gibson assembly into pattB (DGRC #1420) with the *nanos* promoter and 5'UTR.

  To obtain the embryos for ChIP-seq and live embryo imaging in a *zld* knockdown background, we crossed *mat-α-GAL4-VP16 (II)/Cyo; sfGFP-GAF N(III)* flies to *UAS-shRNA-zld (III)* flies and took *mat-α-GAL4-VP16/+ (II); sfGFP-GAF(N)/ UAS-shRNA-zld (III)* females. These females were crossed to their siblings, and their embryos were collected. For controls, *mat-α-GAL4-VP16 (II)/Cyo; sfGFP-GAF N(III)* flies were crossed to $w^{1118}$ flies and embryos from *mat-α-GAL4-VP16/+(II): sfGFP-GAF(N)/+(III)* females crossed to their siblings were collected.

  To obtain embryos for live imaging, hatching rate assays, RNA-seq, ATAC-seq, and ChIP-seq in a GAF knockdown background we crossed *nos-degradFP (II); sfGFP-GAF(N)/TM6c (III)* flies to *His2Av-RFP (II); sfGFP-GAF(N) (III)* flies and selected females that were *nos-degradFP/His2Av-RFP (II); sfGFP-GAF(N) (III)*. These females were crossed to their siblings of the same genotype, and their embryos were collected. Embryos from *His2Av-RFP (II); sfGFP-GAF(N) (III)* females were used as paired controls.

### Cas9-genome engineering

Cas9-mediated genome engineering as described in Hamm et al., 2017 was used to generate the N-terminal and C-terminal super folder Green Fluorescent Protein (sfGFP)-tagged GAF. The double-stranded DNA (dsDNA) donor was created using Gibson assembly (New England BioLabs, Ipswich, MA) with 1 kb homology arms flanking the sfGFP tag and GAF N-terminal or C-terminal open reading frame. sfGFP sequence was placed downstream of the GAF start codon (N-terminal) or just upstream of the stop codon in the fifth GAF exon, coding for the short isoform (C-terminal). Additionally, a 3xP3-DsRed cassette flanked by the long-terminal repeats of PiggyBac transposase was placed in the second GAF intron (N-terminal) or fourth GAF intron (C-terminal) for selection. The guide RNA sequences (N-terminal- TAAACATTAAATCGTCGTGT), (C-terminal- AAATGAATACTCGATTA) were cloned into pBSK under the U63 promoter using inverse PCR. Purified plasmid was injected into embryos of *yw; attP40{nos-Cas9}/CyO* for the N-terminal line and $y^1$ *M{vas-Cas9.RFP-} ZH-2A $w^{1118}$* (BDSC#55821) for the C-terminal line by BestGene Inc Lines were screened for DsRed expression to verify integration. The entire 3xP3-DsRed cassette was cleanly removed using piggyBac transposase, followed by sequence confirmation of precise tag integration.

### Live embryo imaging

Embryos were dechorionated in 50% bleach for 2 min and subsequently mounted in halocarbon 700 oil. Due to the fragility of the GAF^degradFP embryos, embryos used for videos were mounted in halocarbon 700 oil without dechorionation. The living embryos were imaged on a Nikon A1R+ confocal at the University of Wisconsin-Madison Biochemistry Department Optical Core. Nuclear density, based on the number of nuclei/2500 µm2, was used to determine the cycle of pre-gastrulation embryos. Nuclei were marked with His2AV-RFP. Image J (Schindelin et al., 2012) was used for post-acquisition image processing. Videos were acquired at 1 frame every 10 s. Playback rate is seven frames/second.

### Hatching rate assays

A minimum of 50 females and 25 males of the indicated genotypes were allowed to mate for at least 24 hr before lays were taken for hatching rate assays. Embryos were picked from overnight lays and approximately 200 were lined up on a fresh molasses plate. Unhatched embryos were counted 26 hr or more after embryos were selected.

## Immunoblotting

Proteins were transferred to 0.45 µm Immobilon-P PVDF membrane (Millipore, Burlington, MA) in transfer buffer (25 mM Tris, 200 mM Glycine, 20% methanol) for 60 min (75 min for Zld) at 500mA at 4˚C. The membranes were blocked with blotto (2.5% non-fat dry milk, 0.5% BSA, 0.5% NP-40, in TBST) for 30 min at room temperature and then incubated with anti-GFP (1:2000, #ab290) (Abcam, Cambridge, United Kingdom) anti-Zld (1:750) (*Harrison et al., 2010*), or anti-tubulin (DM1A, 1:5000) (Sigma, St. Louis, MO), overnight at 4˚C. The secondary incubation was performed with goat anti-rabbit IgG-HRP conjugate (1:3000) (Bio-Rad, Hercules, CA) or anti-mouse IgG-HRP conjugate (1:3000) (Bio-Rad) for 1 hr at room temperature. Blots were treated with SuperSignal West Pico PLUS chemiluminescent substrate (Thermo Fisher Scientific, Waltham, MA) and visualized using the Azure Biosystems c600 or Kodak/Carestream BioMax Film (VWR, Radnor, PA).

## Fluorescent quantification of nuclei

2–2.5 hr AEL embryos were dechorionated in bleach and subsequently mounted in halocarbon 700 oil. Embryos were imaged on a Nikon Ti-2e Epifluorescent microscope using ×60 magnification. Images were acquired of a single z-plane. Analysis was performed using the Nikon analysis software. Ten circular regions of interest (ROIs) were drawn around individual nuclei and the mean fluorescent intensity of each nucleus was calculated. Ten circular ROIs were drawn in the regions outside of the nuclei to measure the background fluorescent level of the embryo, and the mean fluorescent intensity of the background was calculated. To normalize values to the background fluorescent intensity, the final mean intensity of the nuclei was determined to be the mean fluorescent intensity of the nuclei after subtracting the mean fluorescent intensity of the background. This analysis was performed on images from 27 *GAF-sfGFP(C)* homozygous embryos and 26 *sfGFP-GAF(N)* homozygous embryos.

## Chromatin immunoprecipitation

ChIP was performed as described previously (*Blythe and Wieschaus, 2015*) on: stage 3 and stage five hand selected *GAF-sfGFP(C)* homozygous embryos, stage 3 and stage five hand selected *w^1118^* embryos, 2–2.5 hr AEL hand selected embryos from *mat-α-GAL4-VP16/+ (II); sfGFP-GAF(N)/ UAS-shRNA-zld (III)* females and *mat-α-GAL4-VP16/+ (II); sfGFP-GAF(N)/+ (III)* females, 2–2.5 hr AEL embryos from *nos-degradFP/His2Av-RFP (II); sfGFP-GAF(N) (III)* females, and 2–2.5 hr AEL embryos from *His2Av-RFP (II); sfGFP-GAF(N) (III)* females. For each genotype, two biological replicates were collected. Briefly, 1000 stage 3 embryos or 400–500 stage five embryos were collected, dechorionated in 50% bleach for 3 min, fixed for 15 min in 0.45% formaldehyde and then lysed in 1 mL of RIPA buffer (50 mM Tris-HCl pH 8.0, 0.1% SDS, 1% Triton X-100, 0.5% sodium deoxycholate, and 150 mM NaCl). The fixed chromatin was then sonicated for 20 s 11 times at 20% output and full duty cycle (Branson Sonifier 250). Chromatin was incubated with 6 µg of anti-GFP antibody (Abcam #ab290) or 8 µl of anti-Zld antibody (*Harrison et al., 2010*) overnight at 4˚C, and then bound to 50 µl of Protein A magnetic beads (Dynabeads Protein A, Thermo Fisher Scientific). The purified chromatin was then washed, eluted, and treated with 90 µg of RNaseA (37˚C, for 30 min) and 100 µg of Proteinase K (65˚C, overnight). The DNA was purified using phenol/chloroform extraction and concentrated by ethanol precipitation. Each sample was resuspended in 25 µl of water. Sequencing libraries were made using the NEB Next Ultra II library kit and were sequenced on the Illumina Hi-Seq4000 using 50 bp single-end reads or the Illumina NextSeq 500 using 75 bp single-end reads at the Northwestern Sequencing Core (NUCore).

## ChIP-seq data analysis

ChIP-seq data was aligned to the *Drosophila melanogaster* reference genome (version dm6) using bowtie 2 v2.3.5 (*Langmead and Salzberg, 2012*) with the following non-default parameters: -k 2, --very-sensitive. Aligned reads with a mapping quality < 30 were discarded, as were reads aligning to scaffolds or the mitochondrial genome. To identify regions that were enriched in immunoprecipitated samples relative to input controls, peak calling was performed using MACS v2 (*Zhang et al., 2008*) with the following parameters: -g 1.2e8, --call-summits. To focus analysis on robust, high-quality peaks, we used 100 bp up- and downstream of peak summits, and retained only peaks that were detected in both replicates and overlapped by at least 100 bp. All downstream

analysis focused on these high-quality peaks. Peak calling was also performed for control ChIP samples performed on $w^{1118}$ with the α-GFP antibody. No peaks were called in any of the $w^{1118}$ controls, indicating high specificity of the α-GFP antibody. To compare GAF-binding sites at stage 3, stage 5 and in S2 cells, and to compare GAF, Zld and CLAMP binding, we used GenomicRanges R package (*Lawrence et al., 2013*) to compare different sets of peaks. Peaks overlapping by at least 100 bp were considered to be shared. To control for differences in data processing and analysis between studies, previously published ChIP-seq datasets for Zld (*Harrison et al., 2011*, GSE30757), GAF (*Fuda et al., 2015*, GSE40646), and CLAMP (*Rieder et al., 2019*, GSE133637) were processed in parallel with ChIP-seq data sets generated in this study. DeepTools (*Ramírez et al., 2016*) was used to generate read depth for 10 bp bins across the genome. A z-score was calculated for each 10 bp bin using the mean and standard deviation of read depth across all 10 bp bins. Z-score normalized read depth was used to generate heatmaps and metaplots. De novo motif discovery was performed using MEME-suite (*Bailey et al., 2009*). Genome browser tracks were generated using raw bigWig files with the Gviz package in R (*Hahne and Ivanek, 2016*). Numbers used for all Fisher's exact tests are included in *Supplementary file 4*.

## Spike-in normalization for GAF ChIP in *zld*-RNAi background

Chromatin for GAF ChIP following depletion of Zld was prepared as described above. Prior to addition of the anti-GFP antibody, mouse chromatin prepared from cells expression an H3.3-GFP fusion protein was added to *Drosophila* chromatin at a 1:750 ratio. Following sequencing, reads were aligned to a combined reference genome containing both the *Drosophila* genome (version dm6) and the mouse genome (version mm39). Only reads that could be unambiguously aligned to one of the two reference genomes were retained. To control for any variability in the proportion of mouse chromatin in the input samples, the ratio of percentage of spike in reads in the IP relative to the input were used. A scaling factor was calculated by dividing one by this ratio. Z-score normalized read depth was adjusted by this scaling factor, and the resulting spike-in normalized values were used for heatmaps.

## Analysis of differential binding in GAF- and Zld-depleted embryos

The global decrease in Zld ChIP signal in GAF-depleted embryos (*Figure 4—figure supplement 2C*) caused us to consider technical and biological factors that may have affected these experiments. Global Zld-binding signal may have been lower in the GAF^deGradFP background because GAF^deGradFP embryos die at variable times around the MZT. To attempt account for this variability, we collected embryos from a tight 2–2.5 hr AEL timepoint rather than sorting. Therefore, a portion of these embryos collected for ChIP-seq may have been dead. Additionally, the immunoprecipitation efficiency can vary between experiments and thus might have been lower in the GAF^deGradFP ChIP-seq as compared to the control. Alternatively, as discussed in the Results section, GAF may be broadly required for robust Zld occupancy.

To control for these technical factors, we performed an analysis based on peak rank, in addition to peak intensity. The number of reads aligning within each peak was quantified using featureCounts from the Subread package (v1.6.4) (*Liao et al., 2014*). Peaks were then ranked based on the mean RPKM-normalized read count between replicates, allowing comparison of peak rank between different conditions.

DESeq2 (*Love et al., 2014*) was used to identify potential differential binding sites in a more statistically rigorous way. To control for variable detection of low-intensity peaks, only high-confidence GAF stage 5 peaks identified both in homozygous *GAF-sfGFP(C)* and heterozygous *sfGFP-GAF(N)* embryos (3800 peaks) were analyzed. For Zld ChIP-seq, analysis was restricted to 6003 high-confidence peaks shared between our control dataset and previously published Zld ChIP (*Harrison et al., 2011*). A table of read counts for these peaks, generated by featureCounts as described above, was used as input to DESeq2. Peaks with an adjusted p-value<0.05 and a fold change >2 were considered to be differentially bound. Numbers used for all Fisher's exact tests are included in *Supplementary file 4*.

## Total RNA-seq

A total of 150–200 embryos from *His2Av-RFP/nos-degradFP (II); sfGFP-GAF(N) (III)* and *His2Av-RFP (II); sfGFP-GAF(N) (III)* females were collected from a half hour lay and aged for 2 hr. For each genotype, three biological replicates were collected. Embryos were then picked into Trizol (Invitrogen, Carlsbad, CA) with 200 µg/ml glycogen (Invitrogen). RNA was extracted and RNA-seq libraries were prepared using the Universal RNA-Seq with NuQuant, *Drosophila* AnyDeplete Universal kit (Tecan, Männedorf, Switzerland). Samples were sequenced on the Illumina NextSeq500 using 75 bp single-end reads at the Northwestern Sequencing Core (NUCore).

## RNA-seq analysis

RNA-seq data was aligned to the *Drosophila melanogaster* genome (dm6) using HISAT v2.1.0 (*Kim et al., 2015*). Reads with a mapping quality score <30 were discarded. The number of reads aligning to each gene was quantified using featureCounts, generating a read count table that was used to analyze differential expression with DESeq2. Genes with an adjusted p-value<0.05 and a fold change >2 were considered statistically significant. To identify GAF-target genes, GAF ChIP peaks were assigned to the nearest gene (this study). Zygotically and maternally expressed genes (*Lott et al., 2011*, GSE25180) zygotic gene expression onset (*Li et al., 2014*, GSE58935), Zld-dependent genes (*McDaniel et al., 2019*, GSE121157), CLAMP-dependent genes (*Rieder et al., 2017*, GSE102922), and Zld targets (*Harrison et al., 2011*, GSE30757) were previously defined. Genome browser tracks were generated using bigWigs with the Gviz package in R (*Hahne and Ivanek, 2016*). Numbers used for all Fisher's exact tests are included in *Supplementary file 4*.

## Assay for transposase-accessible chromatin

Embryos from *His2Av-RFP/nos-degradFP (II); sfGFP-GAF(N) (III)* and *His2Av-RFP (II); sfGFP-GAF(N) (III)* females were collected from a half hour lay and aged for 2 hr. Embryos were dechorionated in bleach, mounted in halocarbon 700 oil, and imaged on a Nikon Ti-2e Epifluorescent microscope using 60x magnification. Embryos with nuclei that were not grossly disordered as determined by the His2Av-RFP marker were selected for single embryo ATAC-seq. Six replicates were analyzed for each genotype. Single-embryo ATAC-seq was performed as described previously (*Blythe and Wieschaus, 2016*; *Buenrostro et al., 2013*). Briefly, a single dechorionated embryo was transferred to the detached cap of a 1.5 ml microcentrifuge tube containing 10 µl of ice-cold ATAC lysis buffer (10 mM Tris pH 7.5, 10 mM NaCl, 3 mM MgCl2, 0.1% NP-40). Under a dissecting microscope, a microcapillary tube was used to homogenize the embryo. The cap was placed into a 1.5 ml microcentrifuge tube containing an additional 40 µl of cold lysis buffer. Tubes were centrifuged for 10 min at 500 g at 4˚C. The supernatant was removed, and the resulting nuclear pellet was resuspended in 5 µl buffer TD (Illumina, San Diego, CA) and combined with 2.5 µl H2O and 2.5 µl Tn5 transposase (Tagment DNA Enzyme, Illumina). Tubes were placed at 37˚C for 30 min and the resulting fragmented DNA was purified using the Minelute Cleanup Kit (Qiagen, Hilden, Germany), with elution performed in 10 µl of the provided elution buffer. Libraries were amplified for 12 PCR cycles with unique dual index primers using the NEBNext Hi-Fi 2X PCR Master Mix (New England Biolabs). Amplified libraries were purified using a 1.2X ratio of Axygen magnetic beads (Corning Inc, Corning, NY). Libraries were submitted to the University of Wisconsin-Madison Biotechnology Center for 150 bp, paired-end sequencing on the Illumina NovaSeq 6000.

## ATAC-seq analysis

Adapter sequences were removed from raw sequence reads using NGMerge (*Gaspar, 2018*). ATAC-seq reads were aligned to the *Drosophila melanogaster* (dm6) genome using bowtie2 with the following parameters: `–very-sensitive`, `–no-mixed`, `–no-discordant`, -X 5000, -k 2. Reads with a mapping quality score < 30 were discarded, as were reads aligning to scaffolds or the mitochondrial genome. Analysis was restricted to fragments < 100 bp, which, as described previously, are most likely to originate from nucleosome-free regions (*Buenrostro et al., 2013*). To maximize the sensitivity of peak calling, reads from all replicates of GAF[deGradFP] and control embryos were combined. Peak calling was performed on combined reads using MACS2 with parameters -f BAMPE `–keep-dup` all -g 1.2e8 `–call-summits`. This identified 64,133 accessible regions. To facilitate comparison to published datasets with different sequencing depths, peaks were filtered to include

only those with a fold-enrichment over background > 2.5 (as determined by MACS2). This resulted in 36,571 peaks that were considered for downstream analysis. Because greater sequencing depth can result in the detection of a large number of peaks with a lower level of enrichment over background, this filtering step ensured that peak sets were comparable across all datasets. 201 bp peak regions (100 bp on either side of the peak summit) were used for downstream analysis. Reads aligning within accessible regions were quantified using featureCounts, and differential accessibility analysis was performed using DESeq2 with an adjusted p-value<0.05 and a fold change > 2 as thresholds for differential accessibility. Previously published ATAC-seq datasets (*Hannon et al., 2017*), GSE86966; (*Blythe and Wieschaus, 2016*), GSE83851; (*Nevil et al., 2020*), GSE137075. were reanalyzed in parallel with ATAC-seq data from this study to ensure identical processing of data sets. Heatmaps and metaplots of z score-normalized read depth were generated with DeepTools. MEME-suite was used to for de novo motif discovery for differentially accessible ATAC peaks and to match discovered motifs to previously known motifs from databases. Genome browser tracks were generated using bigWigs with the Gviz package in R (*Hahne and Ivanek, 2016*). Numbers used for all Fisher's exact tests are included in *Supplementary file 4*.

## Acknowledgements

We thank Erica Larschan and Jingyue Duan for helpful discussions. We also thank Peter Lewis, Christine Rushlow, the Bloomington Stock Center, and the *Drosophila* Genome Resource Center for providing reagents and fly lines. We acknowledge the University of Wisconsin-Madison Biochemistry Department Optical Core for access to microscopes for imaging and the University of Wisconsin-Madison Biotechnology Center and the NUSeq Core Facility for sequencing. MMG and TJG were supported by National Institutes of Health (NIH) National Research Service Award T32 GM007215. Experiments were supported by a R01 GM111694 and R35 GM136298 from the National Institutes of Health (NIH) and a Vallee Scholar Award (MMH).

## Additional information

### Funding

| Funder | Grant reference number | Author |
| --- | --- | --- |
| National Institutes of Health | R01GM111694 | Melissa M Harrison |
| National Institutes of Health | R35GM136298 | Melissa M Harrison |
| Vallee Foundation | | Melissa M Harrison |
| National Institutes of Health | T32GM007215 | Marissa M Gaskill Tyler J Gibson |

The funders had no role in study design, data collection and interpretation, or the decision to submit the work for publication.

### Author contributions

Marissa M Gaskill, Conceptualization, Data curation, Formal analysis, Validation, Investigation, Visualization, Methodology, Writing - original draft, Writing - review and editing; Tyler J Gibson, Conceptualization, Data curation, Formal analysis, Validation, Investigation, Visualization, Methodology, Writing - review and editing; Elizabeth D Larson, Formal analysis, Investigation, Writing - review and editing; Melissa M Harrison, Conceptualization, Formal analysis, Supervision, Funding acquisition, Investigation, Visualization, Writing - original draft, Writing - review and editing

### Author ORCIDs

Tyler J Gibson https://orcid.org/0000-0002-2796-2176
Melissa M Harrison https://orcid.org/0000-0002-8228-6836

### Decision letter and Author response

Decision letter https://doi.org/10.7554/eLife.66668.sa1

Author response https://doi.org/10.7554/eLife.66668.sa2

# Additional files

## Supplementary files

• Supplementary file 1. Peaks called in ChIP-seq for GAF-sfGFP (C) in stage 3 and stage 5 hand-sorted embryos and sfGFP-GAF(N) 2–2.5 hr AEL embryos (as indicated in tabs). Chromosome, start and end for each peak are provided as labelled.

• Supplementary file 2. Differentially expressed genes identified by total RNA-seq in GAF$^{deGradFP}$ embryos compared to controls. Columns are defined in the first sheet and data are provided in the other sheet.

• Supplementary file 3. Differential peaks identified in ATAC-seq of GAF$^{deGradFP}$ embryos compared to controls. Columns are defined in the first sheet and data are provided in the other sheet.

• Supplementary file 4. Numbers for statistical analyses performed.

• Transparent reporting form

## Data availability

Sequencing data have been deposited in GEO under accession code GSE152773.

The following dataset was generated:

| Author(s) | Year | Dataset title | Dataset URL | Database and Identifier |
|---|---|---|---|---|
| Gaskill MM, Gibson TJ, Larson ED, Harrison MM | 2020 | GAF is essential for zygotic genome activation and chromatin accessibility in the early *Drosophila* embryo | https://www.ncbi.nlm.nih.gov/geo/query/acc.cgi?acc=GSE152773 | NCBI Gene Expression Omnibus, GSE152773 |

The following previously published datasets were used:

| Author(s) | Year | Dataset title | Dataset URL | Database and Identifier |
|---|---|---|---|---|
| Harrison MM, Li X, Kaplan T, Botchan MR, Eisen MB | 2011 | Zelda binding in the early *Drosophila melanogaster* | https://www.ncbi.nlm.nih.gov/geo/query/acc.cgi?acc=GSE30757 | NCBI Gene Expression Omnibus, GSE30757 |
| Guertin MJ | 2013 | The Genomic Binding Profile of GAGA Element Associated Factor (GAF) in *Drosophila* S2 cells | https://www.ncbi.nlm.nih.gov/geo/query/acc.cgi?acc=GSE40646 | NCBI Gene Expression Omnibus, GSE40646 |
| Rieder L, Jordan W, Larschan E | 2019 | Targeting of the dosage-compensated male X-chromosome during early *Drosophila* development | https://www.ncbi.nlm.nih.gov/geo/query/acc.cgi?acc=GSE133637 | NCBI Gene Expression Omnibus, GSE133637 |
| Lott SE, Eisen MB | 2011 | Non-canonical compensation of zygotic X transcription in *Drosophila melanogaster* development revealed through single embryo RNA-Seq | https://www.ncbi.nlm.nih.gov/geo/query/acc.cgi?acc=GSE25180 | NCBI Gene Expression Omnibus, GSE25180 |
| Li XY, Harrison MM, Kaplan T, Eisen MB | 2014 | Establishment of regions of genomic activity during the *Drosophila* maternal-to-zygotic transition | https://www.ncbi.nlm.nih.gov/geo/query/acc.cgi?acc=GSE58935 | NCBI Gene Expression Omnibus, GSE58935 |
| Rieder LE, Koreski KP, Boltz KA, Kuzu G, Urban JA, Bowman S, Zeidman A, Jordan III WT, Tolstorukov MY, Marzluff WF, Duronio RJ, | 2017 | Histone locus regulation by the *Drosophila* dosage compensation adaptor protein CLAMP | https://www.ncbi.nlm.nih.gov/geo/query/acc.cgi?acc=GSE102922 | NCBI Gene Expression Omnibus, GSE10292 |

Larschan EN

| | | | | |
|---|---|---|---|---|
| Hannon CE, Blythe SA, Wieschaus EF,, Hannon CE, Blythe SA, Wieschaus EF | 2017 | Concentration dependent binding states of the Bicoid Homeodomain Protein | https://www.ncbi.nlm.nih.gov/geo/query/acc.cgi?acc=GSE86966 | NCBI Gene Expression Omnibus, GSE86966 |
| McDaniel SL, Gibson TJ, Schulz KN, Garcia MF, Nevil M, Jain SU, Lewis PW, Zaret KS, Harrison MM | 2019 | Continued activity of the pioneer factor Zelda is required to drive zygotic genome activation | https://www.ncbi.nlm.nih.gov/geo/query/acc.cgi?acc=GSE121157 | NCBI Gene Expression Omnibus, GSE121157 |
| Blythe SA, Wieschaus EF | 2016 | ATAC-seq analysis of chromatin accessibility and nucleosome positioning in *Drosophila melanogaster* precellular blastoderm embryos | https://www.ncbi.nlm.nih.gov/geo/query/acc.cgi?acc=GSE83851 | NCBI Gene Expression Omnibus, GSE83851 |
| Nevil M, Gibson TJ, Bartolutti C, Iyengar A, Harrison MM | 2020 | Developmentally regulated requirement for the conserved transcription factor Grainy head in determining chromatin accessibility | https://www-ncbi-nlm-nih-gov.ezproxy.u-pec.fr/geo/query/acc.cgi?acc=GSE137075 | NCBI Gene Expression Omnibus, GSE137075 |

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
