## [Decision Letter]

**Acceptance summary:**

This paper will be of interest to a broad audience of developmental biologists and molecular biologists in the field of transcriptional control and epigenetics. It evaluates the pioneer factor activity associated with GAGA-Factor during the process of zygotic genome activation. The experiments are rigorously performed and the data analysis supports the conclusions.

**Decision letter after peer review:**

[Editors’ note: the authors submitted for reconsideration following the decision after peer review. What follows is the decision letter after the first round of review.]

Thank you for submitting your work entitled "GAF is essential for zygotic genome activation and chromatin accessibility in the early *Drosophila* embryo" for consideration by *eLife*. Your article has been reviewed by three peer reviewers, and the evaluation has been overseen by a Reviewing Editor and a Senior Editor. The reviewers have opted to remain anonymous. Our decision has been reached after consultation between the reviewers.

This manuscript by Gaskill and Gibson et al. presents compelling evidence that GAGA-factor acts as an additional pioneer factor in the early *Drosophila* embryo. The authors use state of the art genome engineering technology and protein degradation techniques to demonstrate that GAF is essential for ZGA and embryonic viability and is needed for the expression of a set of genes that are known to be bound by GAF.

All the reviewers appreciated the importance and the potential impact of the work, but all raised issues with experimental approaches with so many confounding factors making it difficult to interpret the results with confidence (the detailed critiques can be found below in individual reviews).

In the light of *eLife*'s policy to invite revisions only when revision experiments are unlikely to change major conclusion of the paper, we decided that this manuscript must be rejected at this point. However, we would like to note that all the reviewers are quite enthusiastic about this manuscript, and if you can address major concerns, we would be ready to review the revised manuscript as a new submission, which will be handled by the same set of editors/reviewers.

Reviewer #1:

In this manuscript, Gaskill et al. perform a thorough investigation of the role of GAF as a pioneer transcription factor during zygotic genome activation (ZGA) in *Drosophila* embryos. Using an elegant system for tagging and acute degradation of GAF in embryos, the authors show that: (i) the tagged version of the protein is functional and recapitulates the expected binding properties (Figure 1); (ii) acute proteasomal degradation of the protein prior to ZGA leads to severe defects in embryo viability (Figure 2); (iii) degradation of GAF leads to a significant downregulation of zygotic gene expression at ZGA (Figure 3) independently of Zelda (Figure 4); and, (iv) loss of GAF leads to a significant reduction of chromatin accessibility.

The manuscript is very well written, the data are presented in a logical manner and the findings are supported by the data. The formal identification of GAF as a critical factor for ZGA is important and the results will open up new avenues for investigation of how these molecular mechanisms orchestrate the maternal-to-zygotic transition.

My main criticism of the work is the lack of a biochemical characterisation of GAF binding to nucleosomes to demonstrate a pioneer activity, and further biochemical evidence regarding the level of interaction with Zelda at the protein level. Combined, these experiments might bring a significant level of clarity to some of the results presented in Figure 4 and 5 by elucidating whether GAF is able to bind nucleosomes independently of Zelda.

In addition, I have another three major points:

1) I find the logic regarding the conclusions drawn in Figure 4 difficult to follow. In particular, the authors conclude that "GAF does not broadly depend on Zld for chromatin occupancy in the early embryo". However, Figure 4—figure supplement 1B shows significant differences in the binding intensities upon Zelda RNAi. Therefore, I find it difficult to reconcile these two aspects. I appreciate that the effect might be global, since both Zelda targets and non-Zelda GAF target peaks seem to change, but this needs to be clarified in the manuscript.

2) The same point applies where the authors state: "Zld binding to target loci is largely unperturbed in the absence of GAF". However, Figure 4—figure supplement 1D again shows a significant effect in the binding of Zelda upon GAF removal.

3) I find Figure 5F very difficult to follow and I cannot match the numbers reported in the main text that describe this analysis. This is also the case for Figure 5—figure supplement 1C. The authors should make this part more accessible. In addition, it might be more informative to perform a direct comparison of the ATAC-seq maps generated here and those for Zelda mutants from Hannon et al., 2017. Further analysis on the specific changes in nucleosome occupancy changes after Zelda or GAF depletion measured by these datasets might already help in demonstrating the pioneer activity of GAF.

Reviewer #2:

In the manuscript under review, Gaskill et al., address the longstanding question of the role of GAF in the process of MZT including zygotic genome activation. The question of GAF's role in early development was addressed in the past (Bhat et al., 1996) through conventional genetic means where it was found that elimination of maternal GAF results in a highly pleiotropic, messy, lethal phenotype. Gaskill et al. bring a welcome, timely, and modern approach to revisiting this problem by combining CRISPR, new approaches for generating conditional loss of function (DegradFP), and genomics. In doing so, they recover essentially the same messy pleiotropic loss of function phenotype.

While it would not be impossible to work with such a messy phenotype, in my opinion the authors have not taken sufficient care to ensure that their conclusions are free of possible confounding effects of the GAF loss of function phenotype. As highlighted below, there are several examples where conclusions are over-stated, technical or biological issues remain unaddressed, and -significantly- at least one of the presented experiments is designed so poorly that the rigor of the entire study has to be called into serious doubt (See major issue 1 as well as 3 below).

If done well, the experiments in this study are important to do and the results would be valuable for not only the developmental biology community, but to the broader field of transcriptional regulation and epigenetics. I make the comments below mindful of the difficulties of life in the pandemic, and I have tried to see a way to fix the study without suggesting that significant parts of it be re-done more carefully. Unfortunately, I am not confident that the existing data are suitable for publication as detailed below.

1) Significant lethality of *sfGFP-GAF* (30% hatch rate). While this phenotype is concerning, what is more of an issue is that my interpretation of the westerns in Figure 1—figure supplement 1 suggest that substantially less protein is produced from the N-terminally tagged line as well. This expression problem does not appear to be mentioned in the text, but it definitely should be. What is the quantitative difference between protein levels in these westerns? Does this raise concerns that the “wild-type” samples are not really very wild-type? When does the lethality occur? Specifically, does this effect of the N-terminal GFP-GAF allele affect any of the conclusions?

More concerning is that the authors mixed these two alleles (in different copy numbers as well) for the GAF vs Zld ChIP-seq experiments. As stated in the Materials and methods: these experiments done comparing one versus two copies of the GFP tagged GAF allele, but *also* that the zld RNAi sample used the N-terminal tagged GAF allele (in one copy) and the control used the C-terminal tagged allele (in two copies). As mentioned above, it appears that there is a difference in the expression levels of these two alleles, which is presumably magnified significantly by choosing to compare hets and homozygotes. This experiment could easily have been done on N-terminal hets as controls. This falls well below the bar for acceptable experimental design.

2) Re: conclusion: "maternal GAF is essential for progression through the MZT". The method of knockdown via DegradFP driven by mat-α-GAL4-VP16 begins eliminating GAF in mid-oogenesis. To underscore, this is not an “embryo-specific” knockdown. At the very least, it is a knockdown slightly later than what would be achieved in a germline clone. To the extent that the lethal phenotype has been examined in this manuscript, it remains unclear if it reflects a requirement of GAF specifically for the MZT, or if it reflects a requirement of GAF in some other biological process. There are high degrees of similarity between the DegradFP phenotype and the Trl[13C] phenotype reported in (Bhat '96) where ambiguity about the timing of GAF function also limited insight to when and how GAF is truly required. While the conclusion is not technically incorrect, the implication that this phenotype unambiguously implicates GAF as a regulator of the biological process of MZT (as opposed to oogenesis, genome integrity, chromatin architecture in general, et cetera) is not true.

Also, the GAF loss-of-function phenotype bears absolutely no resemblance to the Zld loss of function phenotype, as the authors further conclude. As the authors are aware, the vast majority of zld germline clones are virtually indistinguishable from wild-type embryos through the cleavage stages, and nearly perfectly phenocopy embryos injected with the Pol 2 inhibitor α amanitin at least at the gross morphological scale. Some minor changes in cell cycle timing occurs in zld mutants, and occasionally an extra mitotic division (also seen with Pol 2). But these cleavages are mostly completed with the fidelity typical of a wild type embryo. In dramatic contrast, GAF loss-of-function results in likely pleiotropic phenotypes that either result in or are caused by errors in mitosis or extensive DNA damage. Importantly, since effects like this are not achieved by inhibiting transcription, this phenotype is likely not due to effects on the ZGA component of MZT. This is consistent with the acute effect of GAF loss of function resulting in disruption of additional unknown maternal processes. But the extent that these disrupted maternal processes specifically impact the MZT, as opposed to other maternally-supplied biological functions, has not been demonstrated in this study.

3) Given the GAF loss of function phenotype: how can the authors be sure that the GAF[degradFP] embryos are at a comparable stage to "wild-type" in their genomic experiments? When gene expression is observed to be GAF-dependent, what steps have the authors taken to ensure that quantitative differences in gene expression stem from loss of GAF and are not confounded by the fact that in a 2-2.5 hr bulk collection that nearly all of the DegradFP embryos are in the process of dying, developing asynchronously, with likely extensive DNA Damage, et cetera? From the Materials and methods section, it seems that for some of the experiments, nothing at all was done. I have a hard time imagining how there is a way around this, since embryos with extensive damaged nuclei cannot be expected to continue transcribing RNA normally, and in fact there is evidence that they very much do not (Iampietro, Dev Cell 2014). Timing is also important here, since many of the genes that are listed as being sensitive to GAF loss of function are just beginning to be expressed, any delay in development would only exacerbate differences between the two experimental samples. For instance, the enrichment for up-regulated maternal genes could either reflect a role for GAF in the process of MZT, in general, or just that the embryos are either significantly delayed or outright die before they have a chance to undergo ZGA. As described above, this does not prove that GAF is involved in the process of MZT. Again, on the basis of these results, it cannot reliably be concluded, that GAF is directly playing a role in the process of ZGA.

4) Looking at the GAF[degradFP] coverage in the bigWig files supplied to GEO, I am strongly concerned that there is a major global difference in the normalized read counts (at least five fold lower peak heights all over the genome, but also with substantially higher background in the DegradFP sample), which should not really be there if the different samples are truly comparable to one another. Is this difference biological or technical? Such huge differences (if they are not just an error in generating the bigWig files) I think would certainly overburden algorithmic normalization in programs like DESeq2. Not to mention that it makes evaluation of the magnitude of differences seen in the ATAC experiments impossible to do through a genome browser, and has significantly colored my interpretation of the results with the DegradFP approach towards the negative (see above). The authors need to address the reason for this difference in magnitude, as well as provide information such as read counts per sample, mapping rates, fraction of mitochondrial reads (for ATAC data). If the issue is due to significant differences in recovered reads between the two genotypes, the question should be asked whether it is a good idea to publish these particular datasets.

Reviewer #3:

This manuscript by Gaskill and Gibson et al. presents compelling evidence that GAGA-factor acts as an additional pioneer factor in the early *Drosophila* embryo. The authors use state of the art genome engineering technology and protein degradation techniques to demonstrate that GAF is essential for ZGA and embryonic viability and is needed for the expression of a set of genes that are known to be bound by GAF. This data provides confirmation that previous results in cell lines translate well in vivo and that the previously established mechanistic role for GAF in opening chromatin and establishing paused Pol II holds true in animals. While this manuscript contains compelling findings and will be of high general interest, the authors' reciprocal ChIP-seq data is not convincing and needs to be repeated.

1) The reciprocal ChIP-seq experiments are simply not convincing in their current form. The authors want to use this data to claim that GAF or Zelda depletion does not impact the binding of the other factor, so that they can say that there is not cooperativity on the "vast majority of target sites". However, they note that there is a global decrease in GAF binding after Zelda depletion and a global decrease in Zelda binding after GAF depletion. They attempt to get around this issue by citing vague technical issues and using rank order comparisons instead, but this analysis is not appropriate in this case. If GAF and Zelda were binding cooperatively, wouldn't one expect there to be a global decrease in the binding of one after depletion of the other, and not necessarily a change in peak rank order? The authors speculate in the Materials and methods section as to why technical reasons could have caused this difference, however, what they should do is perform properly controlled experiments that control for these technical variables so that they can nail this finding with certainty. This is not a trivial issue, as a major conclusion of the paper is that GAF and Zelda act as independent pioneer factors with slightly different activation times in the embryo

a) Why was the zld-RNAi GAF ChIP-seq experiment performed in the *sfGFP-GAF(N)* background and then compared to the *sfGFP-GAF(C)* dataset? This experiment should be repeated with the proper control, GAF ChIP-seq in the *sfGFP-GAF(C)* background. Furthermore, it is concerning that it seems the authors simply performed the zld-RNAi GAF ChIP-seq experiment and then compared back to a previously generated GAF-ChIP-seq dataset; ChIP-seq is an assay subject to a large amount of technical variance and if experiments are to be quantitatively compared as they are in the manuscript, they must be performed at the same time and processed in parallel to minimize this variance. Were the ZLD ChIP-seq experiments and controls performed at the same time and processed in parallel?

b) Why are the authors not utilizing input controls in these ChIP-seq comparisons? Instead of using CPM, why not display fold change over input? This might be a more robust way of dealing with technical differences that lead to global changes as a result of IP efficiency, etc.

---

## [Author Response]

[Editors’ note: the authors resubmitted a revised version of the paper for consideration. What follows is the authors’ response to the first round of review.]

Reviewer #1:In this manuscript, Gaskill et al. perform a thorough investigation of the role of GAF as a pioneer transcription factor during zygotic genome activation (ZGA) in *Drosophila* embryos. Using an elegant system for tagging and acute degradation of GAF in embryos, the authors show that: (i) the tagged version of the protein is functional and recapitulates the expected binding properties (Figure 1); (ii) acute proteasomal degradation of the protein prior to ZGA leads to severe defects in embryo viability (Figure 2); (iii) degradation of GAF leads to a significant downregulation of zygotic gene expression at ZGA (Figure 3) independently of Zelda (Figure 4); and, (iv) loss of GAF leads to a significant reduction of chromatin accessibility.The manuscript is very well written, the data are presented in a logical manner and the findings are supported by the data. The formal identification of GAF as a critical factor for ZGA is important and the results will open up new avenues for investigation of how these molecular mechanisms orchestrate the maternal-to-zygotic transition.My main criticism of the work is the lack of a biochemical characterisation of GAF binding to nucleosomes to demonstrate a pioneer activity, and further biochemical evidence regarding the level of interaction with Zelda at the protein level. Combined, these experiments might bring a significant level of clarity to some of the results presented in Figure 4 and 5 by elucidating whether GAF is able to bind nucleosomes independently of Zelda.

We are pleased that reviewer 1 finds the manuscript clear and that the findings are supported by the data.

Our lab has done several experiments to determine which factors, if any, Zelda is interacting with on the protein level. A graduate student performed IP mass spec, yeast 2-hybrid, and BioID experiments to identify proteins that directly interact with Zelda. Although multiple proteins were identified as potential Zld-interacting factors, we were unable to verify any of these through co-immunoprecipitation followed by immunoblot. We speculate that Zld may interact transiently with a large number of factors, potentially through it’s low-complexity transcription activation domain (Hamm et al., JBC 2015). GAF-interacting partners have been identified using affinity purification coupled with high-throughput mass spectrometry (Lomaev et al., 2017), and Zelda was not identified as a GAF-interacting factor in this study. Based on these previously generated data, we anticipate that if we were to perform co-immunoprecipitation studies with GAF and Zld we would not identify an interaction. We recognize that the biochemical identification of interactions depend on many factors, including purification conditions, making it impossible to rule out protein:protein interactions with a negative result. Therefore, although we cannot definitively state that GAF and Zelda are not interacting on a protein level, previous data suggests it is unlikely. Further supporting independent roles for GAF and Zld in shaping the chromatin landscape, GAF is required for chromatin accessibility in S2 tissue-culture cells, in which Zld is not expressed at detectable levels.

We also agree that nucleosome binding assays with GAF would be interesting and informative, however we believe they are beyond the scope of the current manuscript. A major challenge to this line of experimentation is that we have thus far been unable to purify soluble full length GAF protein in bacterial culture, and therefore anticipate purifying sufficient amounts of full-length GAF will require extensive troubleshooting. Given the difficulty of these experiments we believe it is outside the scope of this study. Although GAF has not yet been shown to directly bind nucleosomes, it has been extensively shown in vitro and in cell culture to promote chromatin accessibility at promoters through the recruitment of chromatin remodelers (Okada and Hirose, 1998; Tsukiyama et al., 1994; Xiao et al., 2001; Tsukiyama and Wu, 1995; Fuda et al., 2015; Judd et al., 2020). Our new analysis includes data demonstrating that GAF binds to regions before accessibility is established. However, we acknowledge that GAF has not been shown to directly bind nucleosomes and therefore it is not proven to have all of the features of a canonical pioneer factor. Therefore, we have removed “pioneer factor” from the title of our manuscript and refer to GAF within the manuscript as a “pioneer-like factor”.

In addition, I have another three major points:1) I find the logic regarding the conclusions drawn in Figure 4 difficult to follow. In particular, the authors conclude that "GAF does not broadly depend on Zld for chromatin occupancy in the early embryo". However, Figure 4—figure supplement 1B shows significant differences in the binding intensities upon Zelda RNAi. Therefore, I find it difficult to reconcile these two aspects. I appreciate that the effect might be global, since both Zelda targets and non-Zelda GAF target peaks seem to change, but this needs to be clarified in the manuscript.

We apologize for the lack of clarity in this section. In response to the concerns of other reviewers, we have repeated these GAF ChIP-seq experiments to better control for any global effects the loss of Zld might have on GAF occupancy. We performed ChIP-seq for *sfGFP-GAF (N)* in a *zld*-RNAi background with paired heterozygous *sfGFP-GAF (N)* embryos as controls. We further included a spike-in control to allow for normalization (chromatin from mouse cells expression GFP-tagged histone H3). In the new GAF ChIP data after z-score and spike-in normalization, we no longer observe a global decrease in GAF binding in the *zld*-RNAi background. When we perform differential binding analysis on these data we again identify relatively few sites (3.3% of total GAF peaks) that are significantly changed. We have extensively rewritten this section of the manuscript and changed Figure 4E and Figure 4—figure supplement 2A, C to incorporate the new data and clarify our conclusions.

2) The same point applies where the authors state: "Zld binding to target loci is largely unperturbed in the absence of GAF". However, Figure 4—figure supplement 1D again shows a significant effect in the binding of Zelda upon GAF removal.

We appreciate the reviewer’s feedback and have rewritten this section to clearly convey our conclusions. We emphasize that although we observe a global decrease in Zld binding upon GAF depletion, a very small subset of Zld binding sites were statistically significantly changed in the absence of GAF (1.7%). Therefore, we conclude that at the majority of Zld binding sites the ability of Zld to bind target loci is not ablated when GAF is depleted. See rewritten section in the comment above.

3) I find Figure 5F very difficult to follow and I cannot match the numbers reported in the main text that describe this analysis. This is also the case for Figure 5—figure supplement 1C. The authors should make this part more accessible. In addition, it might be more informative to perform a direct comparison of the ATAC-seq maps generated here and those for Zelda mutants from Hannon et al., 2017. Further analysis on the specific changes in nucleosome occupancy changes after Zelda or GAF depletion measured by these datasets might already help in demonstrating the pioneer activity of GAF.

We apologize for the lack of clarity in this section. In response to another reviewer’s comments, we performed ATAC-seq on single-embryo replicates. We repeated and expanded our analysis with these new data (new Figure 5 and Figure 6). We now incorporate the ATAC-seq in *zld* mutants from Hannon et al., 2017 in Figure 6. We also investigated when during the MZT regions that require either Zld or GAF for gain accessibility based on data from Blythe and Wieschaus, 2016. We have therefore re-written the Results and Discussion and worked to make these sections more accessible. We have also updated Figure 5, Figure 5—figure supplement1, Figure 6, Figure 6—figure supplement1, and Figure 6—figure supplement 2 to incorporate our new data and analysis.

Reviewer #2:In the manuscript under review, Gaskill et al., address the longstanding question of the role of GAF in the process of MZT including zygotic genome activation. The question of GAF's role in early development was addressed in the past (Bhat et al., 1996) through conventional genetic means where it was found that elimination of maternal GAF results in a highly pleiotropic, messy, lethal phenotype. Gaskill et al. bring a welcome, timely, and modern approach to revisiting this problem by combining CRISPR, new approaches for generating conditional loss of function (DegradFP), and genomics. In doing so, they recover essentially the same messy pleiotropic loss of function phenotype.While it would not be impossible to work with such a messy phenotype, in my opinion the authors have not taken sufficient care to ensure that their conclusions are free of possible confounding effects of the GAF loss of function phenotype. As highlighted below, there are several examples where conclusions are over-stated, technical or biological issues remain unaddressed, and -significantly- at least one of the presented experiments is designed so poorly that the rigor of the entire study has to be called into serious doubt (See major issue 1 as well as 3 below).If done well, the experiments in this study are important to do and the results would be valuable for not only the developmental biology community, but to the broader field of transcriptional regulation and epigenetics. I make the comments below mindful of the difficulties of life in the pandemic, and I have tried to see a way to fix the study without suggesting that significant parts of it be re-done more carefully. Unfortunately, I am not confident that the existing data are suitable for publication as detailed below.

We appreciate that the reviewer notes that the “experiments are important to do and that the results would be valuable” to both the developmental biology and transcription fields and regret that the reviewer did not feel we have “taken sufficient care to ensure our free of confounding effects”. To address these concerns, we have included new ChIP-seq and ATAC-seq data as well as additional data and analysis to further support our conclusions.

1) Significant lethality of sfGFP-GAF (30% hatch rate). While this phenotype is concerning, what is more of an issue is that my interpretation of the westerns in Figure 1—figure supplement 1 suggest that substantially less protein is produced from the N-terminally tagged line as well. This expression problem does not appear to be mentioned in the text, but it definitely should be. What is the quantitative difference between protein levels in these westerns? Does this raise concerns that the “wild-type” samples are not really very wild-type? When does the lethality occur? Specifically, does this effect of the N-terminal GFP-GAF allele affect any of the conclusions?

We appreciate the reviewer’s concerns about the *sfGFP-GAF (N)* line. We were also concerned about any confounding effects, which was the reason we used *sfGFP-GAF(N)* homozygous embryos as our controls for every genomics experiment. By comparing to this control, we will have eliminated from our analysis effects caused by the *sfGFP-GAF(N)* allele alone. We are confident the *sfGFP-GAF(N)* is rescuing essential GAF function, as we have verified the line is homozygous viable and fertile and recapitulates GAF nuclear localization characteristics. Additionally, in this revised manuscript we discuss additional ChIP-seq experiments performed on *sfGFP-GAF(N)* heterozygous embryos. Through comparison of these newly generated data for *sfGFP-GAF(N)* binding to binding sites identified for the C-terminally tagged version (GAF-sfGFP(C)), which has a 66% hatching rate, we identify a high-degree of overlap. 91% of loci identified in GAF-sfGFP(C) embryos are also identified in *sfGFP-GAF(N)* embryos (Figure 4—figure supplement 1B). This demonstrates that the N-terminal sfGFP tag does not alter the ability of GAF to localize to subnuclear domains or to bind specific target loci in the genome. Together, we feel this is strong evidence that, despite the reduced hatching rate, *sfGFP-GAF(N)* recapitulates the majority of GAF function. This analysis is described in the text.

To address the question of protein levels in the two tagged versions of GAF, we performed fluorescent quantification of GFP signal in nuclei of *sfGFP-GAF(N)* and *GAF-sfGFP(C)* homozygous embryos. This strategy allowed us to directly compare single embryos, while our western blots (Figure 1—figure supplement 1D) were performed on bulk collections. This quantification did not identify any significant difference in the fluorescent GFP signal between the two genotypes, suggesting GAF protein levels are not reduced in *sfGFP-GAF(N)* embryos compared to *GAF-sfGFP(C)* embryos. We have included these data in the manuscript in Figure 2—figure supplement 1 and described this experiment in the text.

It is challenging to determine the precise timing of the lethality of the *sfGFP-GAF(N)* embryos as it appears to be variable. We observe that approximately half of the *sfGFP-GAF(N)* homozygous embryos successfully progress to gastrulation. Therefore, a subset of *sfGFP-GAF(N)* embryos die prior to gastrulation, while the remainder die at some point following gastrulation.

More concerning is that the authors mixed these two alleles (in different copy numbers as well) for the GAF vs Zld ChIP-seq experiments. As stated in the Materials and methods: these experiments done comparing one versus two copies of the GFP tagged GAF allele, but *also* that the zld RNAi sample used the N-terminal tagged GAF allele (in one copy) and the control used the C-terminal tagged allele (in two copies). As mentioned above, it appears that there is a difference in the expression levels of these two alleles, which is presumably magnified significantly by choosing to compare hets and homozygotes. This experiment could easily have been done on N-terminal hets as controls. This falls well below the bar for acceptable experimental design.

To address this concern, we performed ChIP-seq for sfGFP-GAF(N) in a *zld*-RNAi background with paired heterozygous *sfGFP-GAF(N)* embryos as controls. Our data for sfGFP-GAF(N) binding from heterozygous embryos identified 6373 GAF peaks. When we compared these to the 4175 GAF peaks identified in the GAF-sfGFP(C) homozygous ChIP-seq dataset, we identified that 91% of GAF-sfGFP(C) peaks identified were maintained in the sfGFP-GAF(N) dataset, and peaks unique to either dataset are among the weakest sites identified (Figure 4—figure supplement 1B). Thus, the location of the sfGFP tag on GAF does not significantly alter the ability of GAF to bind specific genomic loci. To further enable comparison of GAF binding between control and *zld*-RNAi embryos, we included a spike-in control to allow for normalization (chromatin from mouse cells expression GFP-tagged histone H3.3). With the new GAF ChIP data after z-score and spike-in normalization, we no longer observe a global decrease in GAF binding in the *zld*-RNAi background. These new ChIP-seq data support the previous conclusions that GAF remains bound to nearly all the same loci in the presence and absence of Zld. We have updated Figure 4E and Figure 4—figure supplement 2A,C as well as the text to reflect these new data. We were also able to identify 101 GAF-binding sites that were altered upon the depletion of *zld*, and these have been discussed in the text.

2) Re: conclusion: "maternal GAF is essential for progression through the MZT". The method of knockdown via DegradFP driven by mat-α-GAL4-VP16 begins eliminating GAF in mid-oogenesis. To underscore, this is not an “embryo-specific” knockdown. At the very least, it is a knockdown slightly later than what would be achieved in a germline clone. To the extent that the lethal phenotype has been examined in this manuscript, it remains unclear if it reflects a requirement of GAF specifically for the MZT, or if it reflects a requirement of GAF in some other biological process. There are high degrees of similarity between the DegradFP phenotype and the Trl[13C] phenotype reported in (Bhat '96) where ambiguity about the timing of GAF function also limited insight to when and how GAF is truly required. While the conclusion is not technically incorrect, the implication that this phenotype unambiguously implicates GAF as a regulator of the biological process of MZT (as opposed to oogenesis, genome integrity, chromatin architecture in general, et cetera) is not true.

While we highlight that the expression of the degradFP enzyme is driven by the *nanos* promoter and not *mat-α*-GAL4-VP16, we acknowledge the point of the reviewer has raised, and we have changed the language in the manuscript to better reflect this caveat.

We agree that is important to acknowledge potential downstream effects of GAF knockdown at oogenesis. Nonetheless, a number of reasons support our hypothesis that the majority of the effects we identify are the result of GAF knockdown in the embryo:

1) Females in which GAF null germline clones have been generated are rarely able to produce eggs. As reported in Bejarano and Busturia, 2004, of 807 female GAF null germline clones only 23 could produce eggs. Furthermore, most of the eggs produced by these germline clones were unfertilized. These observations suggest that GAF is required during oogenesis for egg production. By contrast, our knockdown system does not show this egg-laying defect, suggesting that the essential functions of GAF in the germline are not inhibited.

2) In Bhat et al., 1996 the authors investigate GAF protein and mRNA levels in the ovary and embryos of wild-type control and *Trl[13C]* hypomorphic GAF mutants. Based on their studies, they conclude that the reduction in *Trl* mRNA deposited in the *Trl[13C]* embryos, and the resulting reduction of GAF protein levels in the embryo, likely produce the nuclear division defects observed. Although they cannot conclusively rule out that the described phenotypes are an indirect effect of GAF knockdown in the germline, they conclude that is unlikely given their observations that GAF protein levels are similar in the ovaries of wild-type controls and *Trl[13C]* mutants. Therefore, the shared phenotypes between our GAF^deGradFP^ embryos and the hypomorphic embryos from *Trl[13C]* mothers in Bhat et al., 1996 are suggestive that GAF knockdown is primarily occurring in the embryo. Nonetheless, our system has advantages over the *Trl[13C]* allele. While Bhat et al. observed variation in phenotype and GAF-expression levels both between and within embryos, our system robustly eliminates GAF protein in every embryo examined and uniformly in all nuclei.

3) Analysis of our genomics data indicates that the effects we observe are the direct consequence of GAF knockdown in the embryo. Our RNA-seq analysis identified only a subset of genes expressed at NC14 that were downregulated in the GAF^deGradFP^ embryos and these were significantly enriched for genes with a proximal GAF-binding site (Figure 3—figure supplement 2D). Further supporting the RNA-seq data and that it reflects the loss of GAF function in the embryo, we identify multiple, zygotically expressed, previously identified GAF-target genes as down-regulated in our RNA-seq analysis, such as *engrailed* and *Ultrabithorax*.

In summary, we agree that we cannot rule out that some of the phenotypic and gene expression differences between the control and GAF^deGradFP^ embryos are due to loss of GAF protein in the female germline. Nonetheless, we feel that the evidence strongly supports that many of the effects we identify are due to protein loss in the embryo.

Also, the GAF loss-of-function phenotype bears absolutely no resemblance to the Zld loss of function phenotype, as the authors further conclude. As the authors are aware, the vast majority of zld germline clones are virtually indistinguishable from wild-type embryos through the cleavage stages, and nearly perfectly phenocopy embryos injected with the Pol 2 inhibitor α amanitin at least at the gross morphological scale. Some minor changes in cell cycle timing occurs in zld mutants, and occasionally an extra mitotic division (also seen with Pol 2). But these cleavages are mostly completed with the fidelity typical of a wild type embryo. In dramatic contrast, GAF loss-of-function results in likely pleiotropic phenotypes that either result in or are caused by errors in mitosis or extensive DNA damage. Importantly, since effects like this are not achieved by inhibiting transcription, this phenotype is likely not due to effects on the ZGA component of MZT. This is consistent with the acute effect of GAF loss of function resulting in disruption of additional unknown maternal processes. But the extent that these disrupted maternal processes specifically impact the MZT, as opposed to other maternally-supplied biological functions, has not been demonstrated in this study.

We agree with reviewer 2 that the phenotypic defects of a subset of GAF knockdown embryos are more dramatic than the effects of maternal *zld* null embryos during NC10-13. Nonetheless, we would like to point out that there are multiple reports of defects in *zld* null embryos prior to NC14. For example, in Liang et al., 2008, nuclear fallout is reported in maternal *zld* null embryos prior to NC14, which we have also observed when imaging maternal *zld*-RNAi embryos. In that same paper, it is described that in early NC14 maternal *zld* null embryos the actin network becomes disorganized and begins to degenerate, leading to highly disorganized nuclei. It is reported in Staudt et al., 2006 that embryos injected with *vielfaltig (*renamed *zld*) dsRNA begin to display anaphase bridges and abnormally condensed chromatin as early as NC10, with these defects becoming more frequent through subsequent nuclear cycles. (Indeed, the gene was originally named *vielfaltig*, meaning versatile or manifold, because of these wide-ranging phenotypic defects.) Therefore, we have edited the text for clarity to reflect that a subset of the GAF knockdown embryos have similar phenotypes to what has been previously published for maternal *zld* knockdown embryos, while a subset of GAF knockdown embryos display more dramatic phenotypes likely due to additional roles of GAF for genome stability.

3) Given the GAF loss of function phenotype: how can the authors be sure that the GAF[degradFP] embryos are at a comparable stage to "wild-type" in their genomic experiments? When gene expression is observed to be GAF-dependent, what steps have the authors taken to ensure that quantitative differences in gene expression stem from loss of GAF and are not confounded by the fact that in a 2-2.5 hr bulk collection that nearly all of the DegradFP embryos are in the process of dying, developing asynchronously, with likely extensive DNA Damage, et cetera? From the Materials and methods section, it seems that for some of the experiments, nothing at all was done. I have a hard time imagining how there is a way around this, since embryos with extensive damaged nuclei cannot be expected to continue transcribing RNA normally, and in fact there is evidence that they very much do not (Iampietro, Dev Cell 2014). Timing is also important here, since many of the genes that are listed as being sensitive to GAF loss of function are just beginning to be expressed, any delay in development would only exacerbate differences between the two experimental samples. For instance, the enrichment for up-regulated maternal genes could either reflect a role for GAF in the process of MZT, in general, or just that the embryos are either significantly delayed or outright die before they have a chance to undergo ZGA. As described above, this does not prove that GAF is involved in the process of MZT. Again, on the basis of these results, it cannot reliably be concluded, that GAF is directly playing a role in the process of ZGA.

We chose to perform a bulk collection of embryos for our RNA-seq experiment precisely because the nuclear defects in the GAF^deGradFP^ embryos makes exactly staging the embryos challenging and because of the issues the reviewer raises given the initiation of gene expression. Given the dramatic changes in gene expression that can occur within minutes during zygotic genome activation, we used tightly timed bulk collection to alleviate potential timing differences that would be exacerbated by single-embryo RNA-seq. To address the concerns above, we analyzed our RNA-seq data by selecting for those genes that initiate expression late in ZGA (late nuclear cycle 14, NC14; the “later” class of genes defined by Li et al., 2014). If expression defects are caused by a failure of the embryos to reach ZGA, we would expect a global decrease in all genes expressed at this timepoint in the GAF^deGradFP^ embryos as compared to control. Instead, our analysis demonstrated that many non-GAF target transcripts activated during late NC14 are expressed at similar levels in GAF^deGradFP^ embryos as compared to control embryos. By contrast, a substantial proportion of genes activated at NC14 with a proximal GAF-binding site (as identified by our ChIP-seq) are downregulated in the GAF^deGradFP^ embryos as compared to controls (Figure 3—figure supplement 2D). Thus, during NC14 direct GAF target transcript expression is affected by GAF knockdown while numerous non-GAF target transcripts are activated normally. This additional analysis suggests our tightly timed bulk collection succeeded in enriching for NC14 embryos and that expression differences identified in our RNA-seq data are not caused by a timing delay in GAF^deGradFP^ embryos compared to controls.

4) Looking at the GAF[degradFP] coverage in the bigWig files supplied to GEO, I am strongly concerned that there is a major global difference in the normalized read counts (at least five fold lower peak heights all over the genome, but also with substantially higher background in the DegradFP sample), which should not really be there if the different samples are truly comparable to one another. Is this difference biological or technical? Such huge differences (if they are not just an error in generating the bigWig files) I think would certainly overburden algorithmic normalization in programs like DESeq2. Not to mention that it makes evaluation of the magnitude of differences seen in the ATAC experiments impossible to do through a genome browser, and has significantly colored my interpretation of the results with the DegradFP approach towards the negative (see above). The authors need to address the reason for this difference in magnitude, as well as provide information such as read counts per sample, mapping rates, fraction of mitochondrial reads (for ATAC data). If the issue is due to significant differences in recovered reads between the two genotypes, the question should be asked whether it is a good idea to publish these particular datasets.

We agree with reviewer 2 that our ATAC-seq data were noisy. Since our lab has had success with single embryo ATAC-seq, we decided to collect single embryos from control and GAF^deGradFP^ embryos from the 2-2.5 hr time point. Single-embryo ATAC-seq assays are not subject to the same timing issues as RNA-seq as the changes in accessibility identified by Blythe and Wieschaus, 2016 over the MZT are less dramatic than the changes in transcript levels during the same time point. Nonetheless, we performed these experiments in six biological replicates to minimize effects of differences in developmental timing. We imaged individual embryos to confirm that they had nuclei that were not grossly disordered using His2av-RFP prior to performing the ATAC-seq. The data from this experiment were much improved, and we have updated and expanded new Figures 5 and 6 using the data from our single embryo ATAC-seq experiment.

We have updated Figure 5, Figure 5—figure supplement1, Figure 6, Figure 6—figure supplement 1, Figure 6—figure supplement 2 with our new data and expanded analysis, as well as the text.

Reviewer #3:This manuscript by Gaskill and Gibson et al. presents compelling evidence that GAGA-factor acts as an additional pioneer factor in the early *Drosophila* embryo. The authors use state of the art genome engineering technology and protein degradation techniques to demonstrate that GAF is essential for ZGA and embryonic viability and is needed for the expression of a set of genes that are known to be bound by GAF. This data provides confirmation that previous results in cell lines translate well in vivo and that the previously established mechanistic role for GAF in opening chromatin and establishing paused Pol II holds true in animals. While this manuscript contains compelling findings and will be of high general interest, the authors' reciprocal ChIP-seq data is not convincing and needs to be repeated.

We appreciate that the reviewer finds the that the manuscript contains “compelling findings and will be of high general interest.” We have repeated the reciprocal ChIP-seq data and re-analyzed these data to address the concerns raised.

1) The reciprocal ChIP-seq experiments are simply not convincing in their current form. The authors want to use this data to claim that GAF or Zelda depletion does not impact the binding of the other factor, so that they can say that there is not cooperativity on the "vast majority of target sites". However, they note that there is a global decrease in GAF binding after Zelda depletion and a global decrease in Zelda binding after GAF depletion. They attempt to get around this issue by citing vague technical issues and using rank order comparisons instead, but this analysis is not appropriate in this case. If GAF and Zelda were binding cooperatively, wouldn't one expect there to be a global decrease in the binding of one after depletion of the other, and not necessarily a change in peak rank order? The authors speculate in the Materials and methods section as to why technical reasons could have caused this difference, however, what they should do is perform properly controlled experiments that control for these technical variables so that they can nail this finding with certainty. This is not a trivial issue, as a major conclusion of the paper is that GAF and Zelda act as independent pioneer factors with slightly different activation times in the embryo.

We agree with the point raised by reviewer 3 and the other reviewers that better controlled experiments eliminating technical variables would bring clarity to the global decrease in GAF and Zld binding observed in the knockdown backgrounds of the other factor. We therefore performed ChIP-seq for *sfGFP-GAF(N)* in a *zld*-RNAi background with paired heterozygous *sfGFP-GAF(N)* embryos as controls. We further included a spike-in control to allow for normalization (chromatin from mouse cells expression GFP-tagged histone H3.3). In our new GAF ChIP-seq data after z-score and spike-in normalization we no longer observe a global decrease in GAF binding in the *zld-*RNAi background. These new data support the previous conclusion that GAF remains bound to the majority of target loci in both the presence and absence of Zld, with only 3.3% of GAF peaks statistically significantly changed in the *zld-*RNAi background. However, we performed additional analysis on the 101 GAF-binding sites that decrease when Zld is lost. We find these sites are enriched for Zld-bound regions that depend on Zld for accessibility. Therefore, at a small subset of GAF-binding sites, pioneering activity of Zld may be required for GAF occupancy. However at the majority of GAF-binding sites GAF does not depend on Zld for occupancy. We used this new GAF ChIP data to generate Figure 4E, Figure 4—figure supplement 2A,C, and we have incorporated textual changes.

We agree with reviewer 3’s assessment that we cannot rule out that the global decrease we observed in Zld binding upon the loss of GAF is biological. However, we expect that if Zld required GAF for occupancy we would see a change in Zld binding in the absence of GAF specifically at regions where the two factors are co-bound. We based this assumption on previously published data on pioneer-factor cooperativity. For example, in Donaghey et al., 2018 the authors found that GATA4 can stabilize FOXA2 binding at a subset of sites where the two factors are bound when they are ectopically expressed in cell culture. Additionally, when OSKM occupancy was investigated during MEF reprogramming it was found that the cooperative binding of O, S, and K was required for targeting of these factors specifically to sites that were co-occupied by the three factors (Chronis et al., 2017). In contrast, the Zld-binding sites that decrease in the absence of GAF are depleted for GAF and Zld co-bound regions when compared to unchanged Zld binding sites. Therefore, at these sites GAF is not directly stabilizing Zld binding or creating local chromatin accessibility. We have clarified the text to more clearly convey this.

Comparisons to previously published reciprocal ChIP-seq data also reinforces our conclusion that neither GAF nor Zld is broadly required for the binding of the other factor to target loci. In Sun et al., 2015 the authors tested the dependency of the transcription factors Dorsal and Zld on the binding of the other factor during the MZT in *Drosophila*. Like us, they used DESeq to call differential binding sites for Dorsal peaks between control and *zld*-RNAi embryos and for Zld peaks in Dorsal mutant and control embryos. They determined that Zld is required for Dorsal binding to a subset of sites, as 19.4% of Dorsal peaks are significantly decreased and 37.8% are significantly changed in the *zld* RNAi background. By contrast, only 2.5% of Zld peaks are significantly changed in the absence of Dorsal. Therefore, the authors conclude that Zld is required for Dorsal binding, but that Dorsal is not required for Zld binding. Our data shows only 3.3% of GAF peaks significantly change upon the loss of Zld and 1.7% of Zld peaks significantly change upon the loss of GAF when we call differential binding with DESeq2 (Figure 4E,F). Therefore, both of our ChIP-seq experiments more closely resemble the impact of Dorsal loss on Zld than the impact of Zld loss on Dorsal. Nevertheless, we have modified the language in our manuscript to make it clear that we cannot rule out an influence of GAF on global Zld binding (Results, Discussion) and to emphasize that there is a small subset of GAF-binding sites that depend on the pioneering activity of Zld for occupancy (Results, Discussion).

a) Why was the zld-RNAi GAF ChIP-seq experiment performed in the sfGFP-GAF(N) background and then compared to the sfGFP-GAF(C) dataset? This experiment should be repeated with the proper control, GAF ChIP-seq in the sfGFP-GAF(C) background. Furthermore, it is concerning that it seems the authors simply performed the zld-RNAi GAF ChIP-seq experiment and then compared back to a previously generated GAF-ChIP-seq dataset; ChIP-seq is an assay subject to a large amount of technical variance and if experiments are to be quantitatively compared as they are in the manuscript, they must be performed at the same time and processed in parallel to minimize this variance. Were the ZLD ChIP-seq experiments and controls performed at the same time and processed in parallel?

As mentioned above, we have repeated the ChIP-seq for sfGFP-GAF(N) in the *zld* RNAi background with paired heterozygous *sfGFP-GAF(N)* controls. In our Zld ChIP-seq experiment, we performed the ChIP-seq for Zld in the GAF^deGradFP^ and paired homozygous sfGFP-GAF(N) controls in parallel.

b) Why are the authors not utilizing input controls in these ChIP-seq comparisons? Instead of using CPM, why not display fold change over input? This might be a more robust way of dealing with technical differences that lead to global changes as a result of IP efficiency, etc.

Previously published ChIP-seq data for Zld at NC14 (Harrison et al., 2011) and GAF in S2 cells (Fuda et al., 2015) that we used for comparisons to our ChIP-seq in the paper did not have publicly available input controls. Therefore, we used CPM normalization for all ChIP-seq data so that we would have a consistent normalization method for ChIP-seq data throughout the paper. Upon revision we have decided to use z-score normalization for all ChIP-seq heatmaps. With our new ChIP-seq data for GAF in a *zld* RNAi background with paired controls using spike-in and z-score normalization we no longer observe a global decrease in GAF signal in the *zld*-RNAi background. However, we continue to observe a global decrease in Zld ChIP signal in GAF knockdown embryos compared to controls using both z-score normalized and input normalization heatmap generation. For consistency, we used z-score normalization throughout.